# Integrated Modelling of Canopy Photosynthesis, Fluorescence, and the Transfer of Energy, Mass and Momentum in the Soil-Plant-Atmosphere Continuum (STEMMUS-SCOPE v1.0.0)

Yunfei Wang[a, b, c, e], Yijian Zeng[c], Lianyu Yu[c], Peiqi Yang[c], Christiaan Van der Tol[c], Qiang Yu[e], Xiaoliang Lü[e], Huanjie Cai[a, b, ※], Zhongbo Su[c, d, ※]

[a] College of Water Resources and Architectural Engineering, Northwest Agriculture and Forestry University, Yangling, China
[b] Institute of Water Saving Agriculture in Arid Regions of China (IWSA), Northwest Agriculture and Forestry University, Yangling, China
[c] Faculty of Geo-Information Science and Earth Observation, University of Twente, Enschede, the Netherlands
[d] Key Laboratory of Subsurface Hydrology and Ecological Effect in Arid Region of Ministry of Education, School of Water and Environment, Chang'an University, Xi'an, China
[e] State Key Laboratory of Soil Erosion and Dryland Farming on the Loess Plateau, Institute of Water and Soil Conservation, Northwest Agriculture and Forestry University, Yangling, China

※ Correspondence: Huanjie Cai (huanjiec@yahoo.com); Zhongbo Su (z.su@utwente.nl)

**Abstract.** Root water uptake by plants is a vital process that influences terrestrial energy, water, and carbon exchanges. In the soil, vegetation, and atmosphere interfaces, root water uptake and solar radiation predominantly regulate the dynamics and health of vegetation growth, which can be remotely monitored by satellites, using the soil-plant relationship proxy – solar-induced chlorophyll fluorescence. However, most current canopy photosynthesis and fluorescence models do not account for root water uptake, which compromises their applications under water stressed conditions. To address this limitation, this study integrated photosynthesis, fluorescence emission, and transfer of energy, mass and momentum in the soil-plant-atmosphere continuum system, via a simplified one-dimensional root growth model and a resistance scheme linking soil, roots, leaves and atmosphere. The coupled model was evaluated with field measurements of maize and grass canopies. The results indicated that the simulation of land surface fluxes was significantly improved by the coupled model, especially when the canopy experienced moderate water stress. This finding highlights the importance of enhanced soil heat and moisture transfer, as well as dynamic root growth, on simulating ecosystem functioning.

**Key words:** SCOPE model; STEMMUS model; Soil-Plant-Atmosphere Continuum (SPAC) system; Root Water Uptake (RWU); Root system growth

## 1. Introduction

Root water uptake (RWU) by plants is a critical process controlling water and energy exchanges between the land surface and the atmosphere and as a result the plant growth. The representation of RWU is an essential component of eco-hydrological

models that simulate terrestrial water, energy and carbon fluxes (Seneviratne et al., 2010; Wang and Smith, 2004). However, most of these models consider the above-ground processes in much greater detail than below-ground processes, and therefore, they have limited ability to represent the dynamic response of plant water uptake to water stress. A particular mechanism of importance for plants to mitigate water stress is the compensatory root water uptake (CRWU) which refers to the process by which water uptake from sparsely rooted but well-watered parts of the root zone compensates for stress in other parts (Jarvis, 2011). The failure to account for compensatory water uptake and the associated hydraulic lift from deep subsoil (Caldwell et al., 1998; Espeleta et al., 2004; Amenu and Kumar, 2007; Fu et al. 2016) can lead to significant uncertainties in simulating the plant growth and corresponding eco-hydrological processes (Seneviratne et al., 2010).

Because the spatial (i.e., one dimensional vertical) pattern of RWU is determined by the spatial distribution of the root system, the knowledge of which is essential for predicting the spatial distribution of water contents and water fluxes in soils. The distribution of roots and their growth are in turn sensitive to various physical, chemical, and biological factors, as well as to soil hydraulic properties that influence the availability of water for plants (Beaudoin et al., 2009). Many attempts have been made in the past to develop root growth models that account for the influence of various environmental factors such as temperature, aeration, soil water availability, and soil compaction. Existing root growth models ranged from complex, three-dimensional root architecture models (Bingham and Wu, 2011; Leitner et al., 2010; Wu et al., 2005) to much simpler root growth models that are implemented within more complex models such as EPIC (Williams et al., 1989) and DSSAT (Robertson et al., 1993). Most of these models reproduce the measured rooting depth very well, but the distribution of new growth root is based on empirical functions rather than biophysical processes (Camargo and Kemanian, 2016) (Table 1).

Modelling RWU requires representation of above and below ground processes, which can be realized considering the flow of water from soil through the plant to the atmosphere (i.e., Soil-Plant-Atmosphere Continuum, SPAC model) (Guo, 1992). The SPAC model represents a good compromise between simplicity (i.e., a small number of tuning parameters) and the ability to capture non-linear responses of RWU (and subsequently the ecosystem functioning) to drought events. Specifically, the SPAC model calculates the CRWU term using the gradient between leaf water potential and soil water potential of each soil layer. The most important parameters in the SPAC model include the leaf water potential, stomatal resistance, and the root resistance. Different from other macroscopic models using the root distribution function, the SPAC model needs explicitly the root length density at each soil layer to calculate the root resistance for each soil layer (Deng et al. 2017). The most practical method for obtaining the root length density is using a root growth model.

On other hand, remote sensing of solar-induced chlorophyll fluorescence (*SIF*) has been deployed to understand and monitor the ecosystem functioning under drought stress using models for vegetation photosynthesis and fluorescence (Zhang et al., 2020; Mohammed et al., 2019; Shan et al., 2019; Zhang et al., 2018). SCOPE (Soil Canopy Observation, Photochemistry, and Energy Fluxes) is such a model, simulating canopy reflectance and fluorescence spectra in the observation directions, as well as photosynthesis, and evapotranspiration as functions of leaf optical properties, canopy structure, and weather variables (Van

der Tol et al. 2009). SCOPE model provides a valuable means to study the link between remote sensing signals and ecosystem functioning, however, it does not consider the water budget in soil and vegetation. As such, there is no explicit parametrization of the effects of soil moisture variations on the photosynthetic or stomatal parameters. Consequently, soil moisture effects are only 'visible' in SCOPE model if the lack of soil moisture affects the optical or thermal remote sensing signals (i.e., during water stress period). The lack of such link between soil moisture availability and remote sensing signals compromises the capacity of SCOPE for simulating and predicting drought events on vegetation functioning.

The change of vegetation optical appearance as a result of soil moisture variations can only explain partially the soil moisture effect on ecosystem functioning (Bayat et al., 2018), which leads to considerably biased estimations of the gross primary productivity *(GPP)* and evapotranspiration (*ET)* in water limited conditions. This presents a challenge for using SCOPE to ecosystems in arid and semi-arid areas, where water availability is the primary limiting factor for vegetation functioning. This challenge becomes even more relevant considering that soil moisture deficit or "ecological drought" is expected to increase in both frequency and severity at nearly all ecosystems around the world (Zhou et al., 2013). Bayat et al. (2019) incorporated the SPAC model into SCOPE to address water stress conditions at a grassland site, but the coupled model neglected the dynamic root distribution at different soil layers and soil moisture serves only as a model input coming from measurements.

In this study, the modelling of above-ground photosynthesis, fluorescence emission, and energy fluxes in the vegetation layer by SCOPE will be fully coupled with a two-phase mass and heat transfer model - STEMMUS model (Simultaneous Transfer of Energy, Mass and Momentum in Unsaturated Soil) (more detailed description of STEMMUS can be found in the *section of methodology and data*), by considering RWU based on a root growth model. The root growth model and the corresponding resistance scheme (from soil, through roots and leaves, to atmosphere) will be integrated for the dynamic modelling of water stress and root system, enabling the seamless modelling of soil-water-plant energy, water and carbon exchanges as well as *SIF*, and thus directly linking the vegetation dynamics (and its optical and thermal appearance) on-process-level to soil moisture variability. The next *section of methodology and data* describes the coupling scheme between SCOPE and STEMMUS models and the data was used to validate the coupled model, followed by the *section of results and discussion* which verifies the coupled STEMMUS-SCOPE model at a maize agroecosystem and a grassland ecosystem located in semi-arid regions and explores the dynamic responses of leaf water potential and root length density to water stress. The summary of this study and the further challenges are addressed in the *section of conclusions*.

**Table 1. Comparison of LSMs and crop models in terms of sink term calculation of soil water balance.**

| | Model | Sink term calculation of soil water balance | Root water uptake process | | |
| --- | --- | --- | --- | --- | --- |
| | | | Hydraulic redistribution (Richards and Caldwell, 1987) | Compensatory uptake (Jarvis, 2011) | Root distribution |
| LSMs | CLM5.0 | Root length density of each soil layer and water stress is applied by the hydraulic conductance model (Lawrence et al. (2020) | Extreme case of CRWU | Following Darcy's Law for porous media flow equations | Empirical function depends on the plant functional type |
| | CLM4.5 | Actual transpiration, root fraction of each soil layer and soil integral soil water availability (Fu et al., 2016) | The Ryel et al. (2002) function | Not considered | Empirical function |
| | CLM4.0 | Actual transpiration, root fraction of each soil layer and soil integral soil water availability (Couvreur et al. 2012, Sulis et al., 2019) | HRWU scheme (RWU model based on hydraulic architecture) | HRWU scheme | Empirical function |
| | CLM3 & IBIS2 | Actual transpiration, physical root distribution and the water availability in each layer (Zheng and Wang, 2007) | The Ryel et al. (2002) function | Dynamic root water uptake | Empirical function |
| | CoLM | Potential transpiration, root fraction in each layer and water stress factor (Zhu et al., 2017) | The Ryel et al. (2002) and the Amenu and Kumar (2007) function | Empirical approach with a compensatory factor | Empirical function |
| | JULES | Potential transpiration, root fraction of each soil layer and a weighted water stress in each layer (Eller et al., 2020) | Not considered | Not considered | Exponential distribution with depth |
| | Noah-MP | Based on the gradient in water potentials between root and soil, and root surface area (Niu et al., 2010) | Extreme case of CRWU | Following Darcy's law for porous media flow equations | Process-based 1D root surface area growth model |
| | CABLE | Based on the gradient in water potentials between the leaf, stem, and the weighted average of the soil (De Kauwe et al., 2020) | Extreme case of CRWU | Following Darcy's law for porous media flow equations | Empirical function |
| Crop Models | APSIM | Potential transpiration and water supply factor, but neglect root distribution (Keating et al., 2003) | Not considered | Not considered | Empirical function |
| | CropSyst | Difference in water potential between the soil and the leaf, and a total soil–root–shoot conductance (Stöckle et al., 2003) | Not considered | Considered by the leaf and soil water potential | Linear decrease in soils with No limitations to root exploration |
| | DSSAT | Water uptake per unit of root length is computed as an exponential function, and the actual RWU is the minimum of potential transpiration and the maximum capacity of root water uptake (Jones et al., 2003) | Not considered | Water uptake per unit of root length as a function of soil moisture | Using an empirical function |
| | EPIC | EPIC assumes that water is used preferentially from the top layers, and the potential water supply rate decreases exponentially downward (Williams et al.,2014) | Not considered | Not considered | Not considered |
| | SWAP | Based on the potential transpiration, root fraction and an empiric stress factor relationship (van Dam, 2000) | Not considered | Based on soil water potential | Function of relative rooting depth |
| | WOFOST | The simplest one, it calculates water uptake as a function of the rooting depth and the water available in that rooting depth without regard to the soil water distribution with depth (Supit et al., 1994) | Not considered | Not considered | Empirical function |
| | SPACSYS | According to empirical root length density distribution in a soil layer, potential transpiration and soil moisture (Wu et al., 2005) | Not considered | Not considered | 1D (empirical function) or 3D root system (process based) |
| | STICS | Based on the potential transpiration, root fraction, and soil water distribution, but not process based (Beaudoin et al., 2009) | Not considered | Not considered | 1D root length density profile |

## 2. Methodology and Data

### 2.1. SCOPE and SCOPE_SM Models

SCOPE is a radiative transfer and energy balance model (Van der Tol et al. 2009). It simulates the transfer of optical, thermal, and fluorescent radiation in the vegetation canopy and computes *ET* by using an energy balance routine. SCOPE includes a radiative transfer module for incident solar and sky radiation to calculate the top of canopy outgoing radiation spectrum, net radiation and absorbed photosynthetically active radiation (*aPAR*), a radiative transfer module for thermal radiation emitted by soil and vegetation to calculate the top of canopy outgoing thermal radiation and net radiation, an energy balance module for latent heat, sensible heat and soil heat flux, and a radiative module for chlorophyll fluorescence to calculate the top of canopy *SIF* (the observation zenith angle was set as zero degree in this study).

Compared to other radiative transfer models which simplify the radiative transfer processes based on Beer's law, SCOPE has well-developed radiative transfer modules which consider the various leaf orientation and multiple scattering. SCOPE can provide detailed information about net radiation of every leaf within the canopy. Furthermore, SCOPE incorporates an energy balance model which predicts not only the temperature of leaf but also the temperature of soil surface temperature (i.e., a vital boundary condition needed by STEMMUS). In the original SCOPE, soil is treated in a very simple way with several empirical functions describing the ground heat storage. Later, Bayat et al. (2019) extended the SCOPE model by including the moisture effects on the vegetation canopy, which results in the SCOPE_SM model. This model takes soil moisture as input and predicts the effects on several processes of vegetation canopy by using the SPAC concept. Appendix A.1 lists the main equations of calculating water stress factor within SCOPE (Bayat et al. 2019), and the detailed formulation of SCOPE is referred to Van der Tol et al. (2009).

SCOPE_SM provides the basic framework to couple SCOPE with a soil process model. However, both SCOPE and SCOPE_SM ignored the soil heat and mass transfer processes and the dynamics of root growth. This can be overcome by introducing the STEMMUS model.

### 2.2. STEMMUS Model

STEMMUS model is a two-phase mass and heat transfer model with explicit consideration of the coupled liquid, vapor, dry air and heat transfer in unsaturated soil (Zeng et al. 2011a,b; Zeng and Su, 2013; Yu et al. 2018). STEMMUS provides a comprehensive description of water and heat transfer in the unsaturated soil, which can compensate what is currently neglected in SCOPE. In STEMMUS, the soil layers can be set flexible which was better than previous SPAC model only considered the whole root zone soil water content as fixed layers (Williams et al., 1996). The water and heat transfer processes are vital for vegetation phenology development as well as freeze-thaw processes. The boundary condition needed by STEMMUS includes surface soil temperature, which is the output of SCOPE. In addition, STEMMUS already contained an empirical equation to

calculate root water uptake and a simplified root growth module to calculate root fraction profile. As such, STEMMUS has an ideal model structure to be coupled with SCOPE. The main governing equations of STEMMUS are listed in Appendix A.2.

## 2.3 Dynamic Root Growth and Root Water Uptake

To obtain the root resistance of each soil layer, we incorporated a root growth module to simulate the root length density profile (see Appendix A.3). The simulation of root growth refers to the root growth module in the INRA STICS crop growth model (Beaudoin et al., 2009), which includes the calculations of root front growth and root length growth. The root front growth is a function of temperature, with the depth of the root front beginning at the sowing depth for sown crops and at an initial value of transplanted crops or perennial crops (Beaudoin et al., 2009). The root length growth is calculated in each soil layer, considering the net assimilation rate and the allocation fraction of net assimilation to root, which is in turn a function of leaf area index (LAI) and root zone water content (Krinner et al. 2005). The root length density profile is then used to calculate the root resistance to water flow radially across the roots, soil hydraulic resistance, and plant axial resistance to flow from the soil to the leaves (see Appendix A.4).

## 2.4 STEMMUS-SCOPE v1.0.0 Coupling

The coupling starts with an initial soil moisture (SM) profile simulated by STEMMUS, which enables the calculation of the water stress factor as a reduction factor of the maximum carboxylation rate ($V_{cmax}$), SCOPE v1.73 is then used to calculate net photosynthesis ($A_n$) or gross primary productivity (GPP), soil respiration (Rs), energy fluxes (Rn, LE, H and G), transpiration (T) and SIF, which is passed to STEMMUS as the root water uptake (RWU). Then, the gross primary production (GPP) can be calculated based on $A_n$. Surface soil moisture is also used in calculating soil surface resistance and then calculating soil evaporation (E). Furthermore, SCOPE can calculate soil surface temperature ($T_{s0}$) based on energy balance, which was subsequently used as the top boundary condition of STEMMUS, and leaf water potential (LWP), which is a parameter to reflect plant water status, can be calculated through iteration. Based on RWU, STEMMUS calculates the soil moisture in each layer at the end of the time step, and the new soil moisture profile will be the soil moisture at the beginning of next time step, which is repeated as such till the end of simulation period. The time-step of STEMMUS-SCOPE is flexible and the time step used in this study was half hour. Figure 1 shows the coupling scheme of STEMMUS and SCOPE, and Table B.1 shows all the parameter values used in this study.

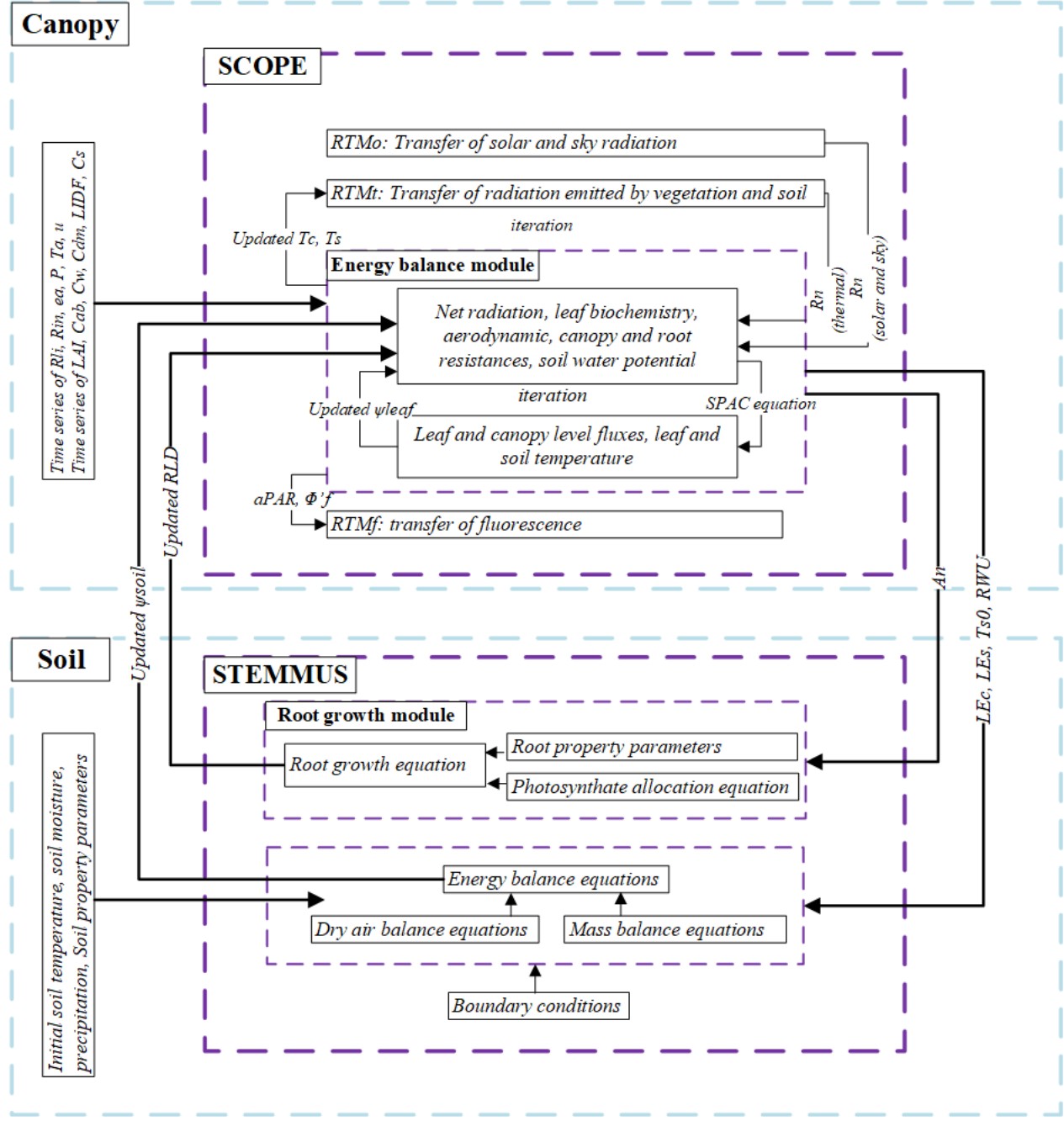

150

**Figure 1. The coupling scheme of STEMMUS-SCOPE. The explanations of the symbols were the same as in Table B.1.**

## 2.5. Evapotranspiration partitioning

Most studies in partitioning evapotranspiration (*ET*) use sap flow and micro lysimeter data from in-situ measurements. In this study, we used a simple and practical method to separate evaporation (*E*) and transpiration (*T*) proposed by Zhou et al. (2016). Although the behaviour of plant stomata is influenced by environmental factors, the potential water use efficiency (*uWUE$_p$*, g C hPa$^{0.5}$/kg H$_2$O) at stomatal scale in the ecosystem with a homogeneous underlying surface is assumed to be nearly constant, and variations of actual *uWUE* (g C hPa$^{0.5}$/kg H$_2$O) can be attributed to the soil evaporation (Zhou et al., 2016). Thus, the method can be used to estimate *T* and E with the quantities of *ET*, *uWUE* and *uWUE$_p$*. Another assumption of this method is that the ecosystem *T* equal to *ET* at some growth stages, so *uWUE$_p$* can be estimated using the upper bound of the ratio of $GPP\sqrt{VPD}$ to *ET* (here VPD refers to vapor pressure deficit) (Zhou et al., 2014; Zhou et al., 2016).

Zhou et al. (2016) used the 95[th] quantile regression between $GPP\sqrt{VPD}$ and *ET* to estimate *uWUE$_p$*, and showed that the 95[th] quantile regression for *uWUE$_p$* at flux tower sites was consistent with the *uWUE* derived at the leaf scale for different ecosystems. In addition, the variability of seasonal and interannual *uWUE$_p$* was relatively small for a homogeneous canopy. Therefore, the calculations of *uWUE$_p$*, *uWUE*, and *T* at the ecosystem scale were as follows:

$$uWUE_p = \frac{GPP\sqrt{VPD}}{T} \tag{1}$$

$$uWUE = \frac{GPP\sqrt{VPD}}{ET} \tag{2}$$

$$\frac{T}{ET} = \frac{uWUE}{uWUE_p} \tag{3}$$

The calculation of *VPD* was based on air temperature and relative humidity data, and the method of gap-filling was the Marginal Distribution Sampling *(MDS)* method proposed by Reichstein et al. (2005). To calculate *GPP*, the complete series of net ecosystem exchange *(NEE)* was partitioned into gross primary production *(GPP)* and respiration *(Re)* using the method proposed by Reichstein et al. (2005). Finally, *ET* was calculated using the latent heat flux and air temperature. Based on *GPP*, *ET* and *VPD* data, *T* can be calculated using the method proposed by Zhou et al. (2016).

## 2.6. Study site and data description

To evaluate the performance of STEMMUS-SCOPE in modelling ecohydrological processes, simulation was conducted to compare STEMMUS-SCOPE with SCOPE, SCOPE_SM, and STEMMUS using observations over a C4 cropland (Summer-maize: from 11 June to 10 October 2017) at the Yangling station (34°17′ N, 108°04′ E, 521 m a.s.l.) and a C3 grassland at the Vaira Ranch (US-Var) Fluxnet site (38°25′ N, 120°57′ W, 129 m a.s.l.) (Annual grasses: from 1 June to 8 August 2004). The seasonal variation of precipitation, irrigation, and *SM* for these two sites were presented in Figure 2. And the differences of soil surface resistance, water stress factor (*WSF*), *ET*, photosynthesis, soil surface temperature (*T$_s$0*), root water uptake (RWU)

and leaf water potential (*LWP*) between these four models were presented in Table 2. In this study, the LAI data of Vaira
Ranch (US-Var) Fluxnet site was from MODIS 8-daily LAI product instead of the field measured LAI used by Bayat et al.
(2019). For the soil water content used by SCOPE_SM, the averaged root zone soil moisture was used for Yangling station
and the soil moisture at 10 cm depth was used for Vaira Ranch site. More detailed descriptions of these sites and data can refer
to Wang et al. (2019; 2020a) and Bayat et al. (2018; 2019).

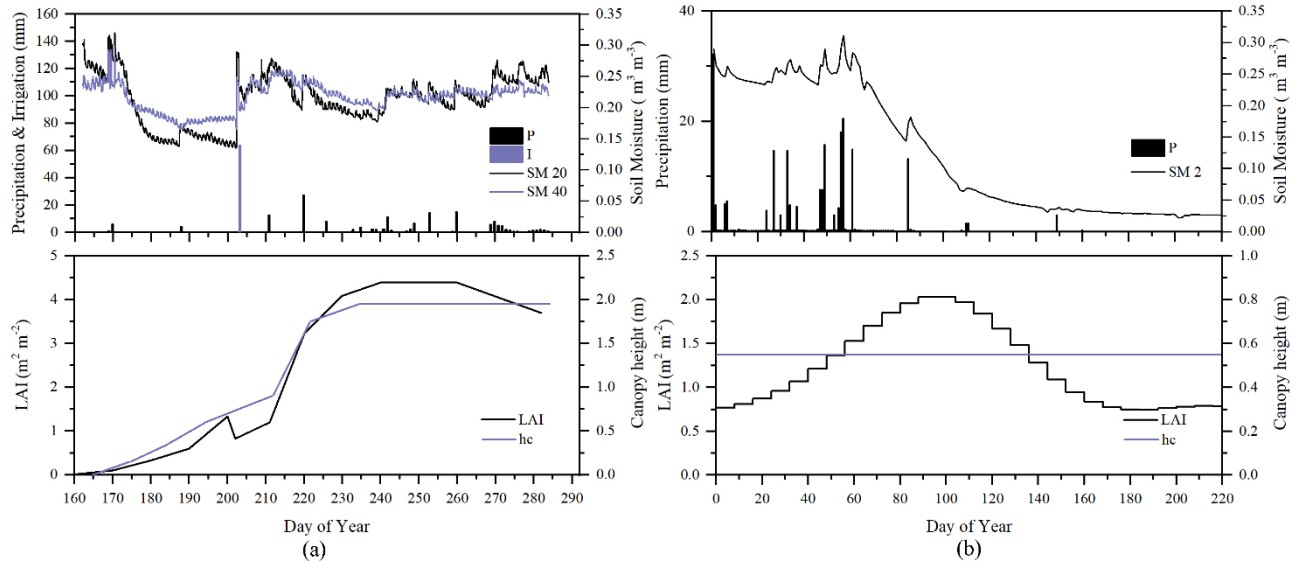

(a)                                    (b)

**Figure 2 Seasonal variation of precipitation (P), irrigation (I), soil moisture at 2cm (SM 2), 20 cm (SM 20), 40 cm depth (SM 40), Leaf area index (LAI), and canopy height ($h_c$): (a) Maize cropland at Yangling station; (b) Grassland at Vaira Ranch (US-Var) Fluxnet site.**

**Table 2. Main differences among SCOPE, SCOPE_SM, STEMMUS, and STEMMUS-SCOPE.**

| | SCOPE | SCOPE_SM | STEMMUS | STEMMUS-SCOPE |
|---|---|---|---|---|
| Source | Van der Tol et al. (2009) | Bayat et al. (2019) | Zeng et al. (2013) | This study |
| *Soil surface resistance* calculation | Set *SM* as constant or field measured surface *SM* | Field measured surface *SM* | Simulated surface *SM* by itself | Simulated surface *SM* by itself |
| *WSF* calculation | Set *SM* as constant | Field measured *SM* | Simulated *SM* by itself | Simulated *SM* by itself |
| *ET* calculation | Process based (Analogy with Ohm's law) | Process based (Analogy with Ohm's law) | Penman–Monteith model or FAO dual crop coefficient method | Process based (Analogy with Ohm's law) |
| Photosynthesis | Farquhar and Collatz model | Farquhar and Collatz model | Absent | Farquhar and Collatz model |
| Radiation transfer | SAIL4 model | SAIL4 model | Based on Beer's law | SAIL4 model |
| $T_s0$ | Simulated by itself | Simulated by itself | Field measured | Simulated by itself |
| *RWU* calculation | Absent | Absent | Based on potential $T$, root fraction, and soil moisture profile | Based on leaf and soil water potential |
| *LWP calculation* | Absent | Calculated by iteration | Absent | Calculated by iteration |
| Root growth | Absent | Absent | Empirical model | Process-based model |

## 2.7. Performance Metrics

The metrics used to evaluate the performance of coupled STEMMUS-SCOPE model include: (1) Root Mean Squared Error (*RMSE*); (2) coefficient of determination ($R^2$); and (3) the index of agreement (*d*). They are calculated as:

$$RMSE = \sqrt{\frac{1}{n}\sum_{i=1}^{n}(P_i - O_i)^2} \tag{4}$$

$$R^2 = \frac{[\sum_{i=1}^{n}(P_i - \bar{P})(O_i - \bar{O})]^2}{\sum_{i=1}^{n}(P_i - \bar{P})^2 \sum_{i=1}^{n}(O_i - \bar{O})^2} \tag{5}$$

$$d = 1 - \frac{\sum_{i=1}^{n}(P_i - O_i)^2}{\sum_{i=1}^{n}(|P_i - \bar{O}| + |O_i - \bar{O}|)^2} \tag{6}$$

where $P_i$ is the *i*th predicted value, $O_i$ is the *i*th observed value, $\bar{O}$ is the average of observed values, and *n* is the number of samples.

## 3. Results and discussion

### 3.1. Soil moisture modelling

As the soil moisture profile was not available in US-Var site, the comparisons of simulated soil moisture (*SM*) at Yangling station using STEMMUS and STEMMUS-SCOPE and observed ones are presented in Figure 3. For the simulation of soil moisture at 20 cm, the RMSE value was 0.023 and 0.021, and d value was 0.90 and 0.91, for STEMMUS and STEMMUS-SCOPE respectively. For the simulation of soil moisture at 40cm, the RMSE value was 0.017 and 0.021 and d value was 0.83

and 0.74, respectively. The simulated soil moisture at 20 cm depth agreed with the observed values in terms of seasonal pattern. Although slight overestimation occurred at initial and late stages, the dynamics in soil moisture resulted from precipitation or irrigation were well captured. Per the nature of the two models, the coupling of SCOPE with STEMMUS is not expected to improve the simulation of soil moisture. However, compared to SCOPE_SM, which used soil moisture measurements as inputs, the coupled STEMMUS-SCOPE improves the simulation of soil moisture dynamics as measured. The deviation between the model simulations and the measurements can be attributed to the following two potential reasons. First, the field observation has errors to a certain extent and the soil moisture sensors may be not well calibrated. Second, in this simulation, we assumed that the soil texture was homogeneous in the vertical profile, whereas the soil properties (e.g. soil bulk density and saturated hydraulic conductivity) may vary with depth in reality, and at different growth stages due to field management practices. For example, the soil bulk density at 40 cm was much higher than that at 20 cm due to the mechanical tillage, especially in the early stage.

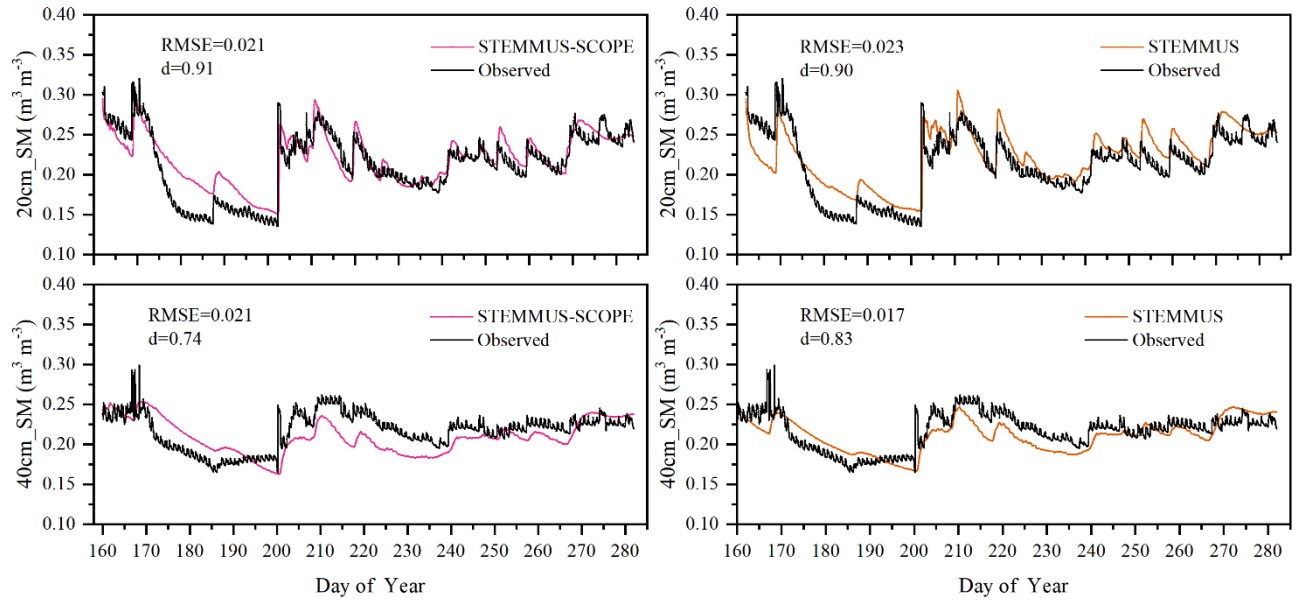

**Figure 3 Comparison of modeled and observed soil moisture at 20 cm (20 cm_SM) and 40 cm depth (40 cm_SM) for the maize cropland at Yangling station.**

### 3.2. Soil temperature modelling

Similar to soil moisture, only simulated soil temperatures ($T_s$) at Yangling site by STEMMUS and STEMMUS- SCOPE at 20 cm and 40 cm depth are shown in Figure 4. In general, both two models can capture the dynamics of soil temperature well. For the simulation of temperature at 20 cm, the RMSE value was 2.56 °C and 2.58 °C, and d value was 0.92 and 0.92, for STEMMUS and STEMMUS-SCOPE respectively. For the simulation of temperature at 40cm, the RMSE value was 2.06 °C

and 2.07 °C, and d value was 0.93 and 0.93, respectively. These results indicate that both models can simulate well soil temperature. However, there also exist some differences between simulation and observation. The largest difference occurred in DOY 202, when the field was irrigated with the flooding irrigation method. This irrigation activity may lead to the boundary condition errors (i.e., for soil surface temperature), which cannot be estimated well enough (e.g., there is no monitoring of water temperature from the irrigation). Meanwhile, the measurement may also have some errors in this period. The fact for the observed soil temperature at 20 cm and 40 cm decreasing to almost the same level at the same time indicates a potential pathway for preferential flow in the field (see precipitation/irrigation at DOY 202 in Figure 2), and the sensors captured this phenomenon. Nevertheless, the model captures the soil temperature dynamics.

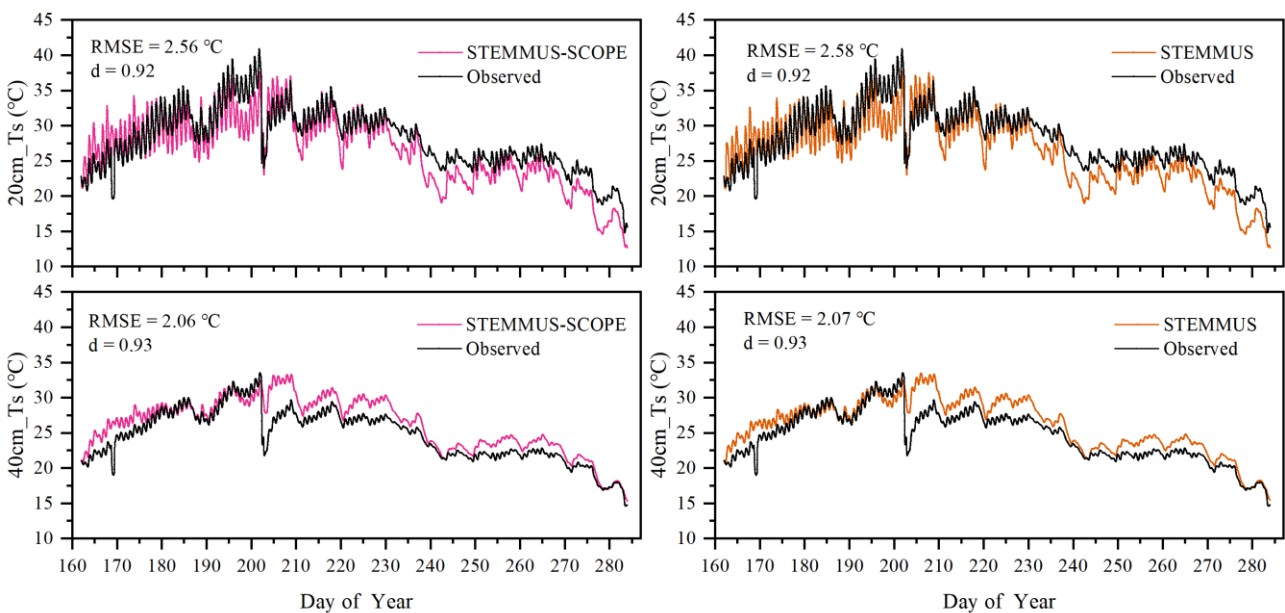

**Figure 4 Comparison of observed and modeled soil temperature at 20 cm (20 cm_Ts) and 40 cm depth (40 cm_Ts) for the maize cropland at Yangling station.**

### 3.3. Energy balance modelling

A comparison of the modeled and observed half-hourly net radiation ($Rn$), sensible heat flux ($H$), latent heat flux ($LE$), and soil heat flux ($G$) using SCOPE, SCOPE_SM, and STEMMUS-SCOPE are presented in Figure 5 (STEMMUS uses $Rn$ as driving data and therefore is not included in the comparison). For net radiation and soil heat flux, the simulations of all three models show good agreements with observations, and the coefficient of determination ($R^2$) for SCOPE, SCOPE_SM and STEMMUS-SCOPE was 0.99, 1.00, and 0.99, respectively. For soil heat flux, the $R^2$ for SCOPE, SCOPE_SM and STEMMUS-SCOPE was 0.81, 0.79, and 0.80, respectively. For latent heat flux, STEMMUS-SCOPE has a better performance than SCOPE and SCOPE_SM, and the $R^2$ for SCOPE, SCOPE_SM and STEMMUS-SCOPE was 0.82, 0.84, and 0.85,

respectively. Furthermore, STEMMUS-SCOPE and SCOPE_SM have a similar performance in the simulation of sensible heat flux, which were better than the performance of SCOPE, and the $R^2$ for SCOPE, SCOPE_SM and STEMMUS-SCOPE was 0.70, 0.75, and 0.74, respectively.

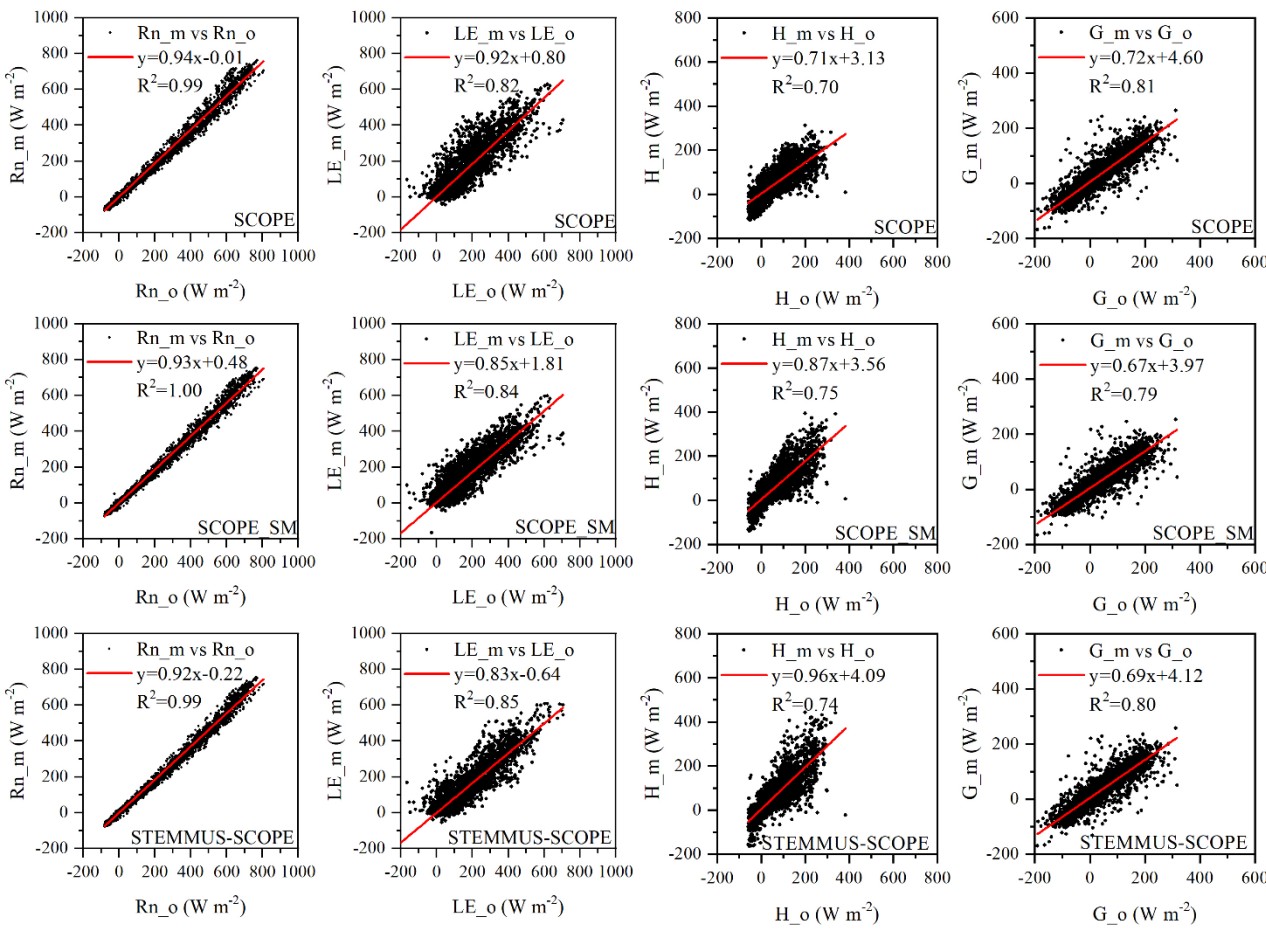

**Figure 5 Comparison of modelled and observed half-hourly net radiation (Rn), latent heat (LE), sensible heat (H) and soil heat flux (G) by SCOPE, SCOPE_SM and STEMMUS-SCOPE at Yangling station. Subscripts '_m' and '_o' in each plot indicates modeled and observed quantities, respectively. The regression line is indicated in red color with the corresponding regression equation and the $R^2$.**

**3.4. Daily ET, T and E modelling**

Simulated daily evapotranspiration (ET) results by SCOPE, SCOPE_SM, STEMMUS and STEMMUS-SCOPE are presented in Figure 6. For the Yangling station, the $R^2$ by SCOPE, SCOPE_SM, STEMMUS and STEMMUS-SCOPE was 0.76, 0.82, 0.80 and 0.81, and the $RMSE$ was 0.84, 0.69, 0.76, and 0.74 mm day$^{-1}$, respectively. For the US-Var station, the $R^2$ by SCOPE, SCOPE_SM, STEMMUS and STEMMUS-SCOPE was 0.10, 0.66, 0.84 and 0.89 and the $RMSE$ was 1.83, 0.63, 0.40, and 0.34

255  mm day$^{-1}$, respectively. For the ET simulation by SCOPE, there were large differences between simulations and observations

when the vegetation suffered water stress. For SCOPE_SM, STEMMUS and STEMMUS-SCOPE, because of including the dynamics of soil moisture, the simulated ET were closer to observations when the vegetation experienced water stress. It indicates that STEMMUS-SCOPE, STEMMUS and SCOPE_SM can predict ET with a relatively higher accuracy, especially when the maize was under water stress (DOY 183 to 202 at Yangling station and DOY 90 to 220 at US-Var site), and

STEMMUS-SCOPE and SCOPE_SM performed similarly well. It is noteworthy that although STEMMUS has considered the effect of soil moisture on ET, the accuracy of STEMMUS was lower than the coupled model (see Figure 6). The possible reason is the better representation of transpiration in SCOPE model (see Figure 7), which separates the canopy into 60 layers, while STEMMUS only treats the canopy as one layer. Besides, the coupled model performed better at grassland than at maize cropland. The reason is that the grassland simulation used the dynamic $V_{cmax}$ data while the maize simulation used a constant

$V_{cmax}$                                                                                                                                                                                                                                                 data.

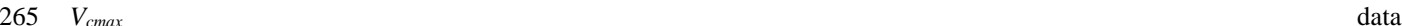
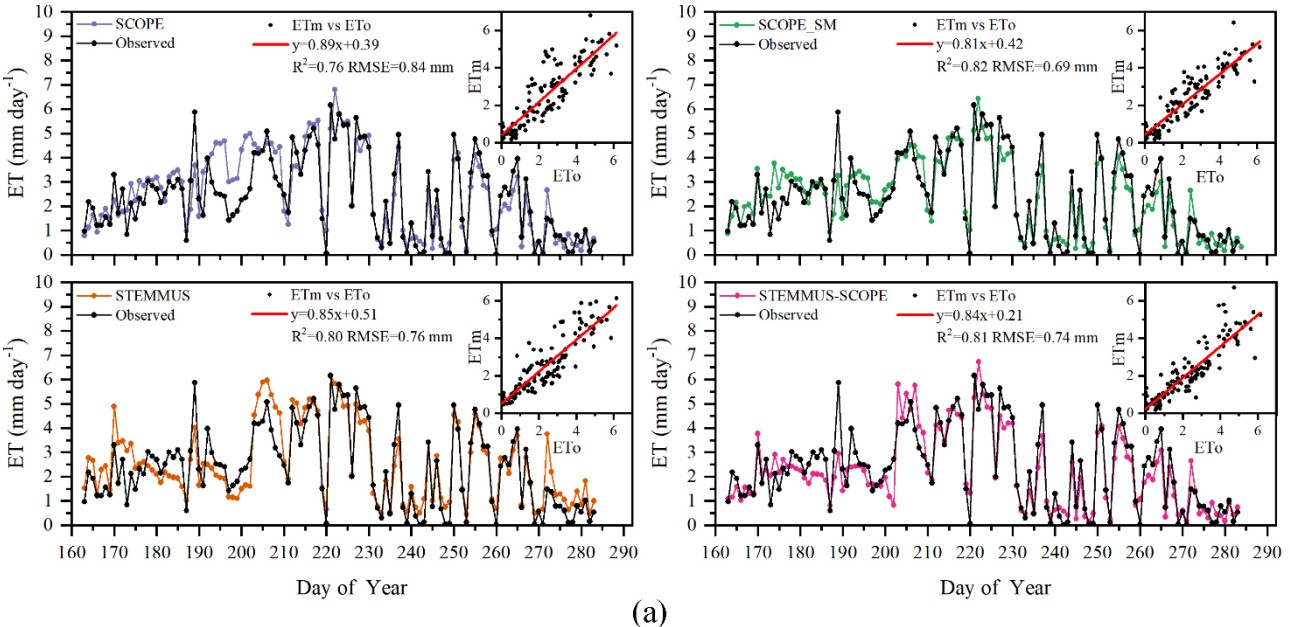

(a)

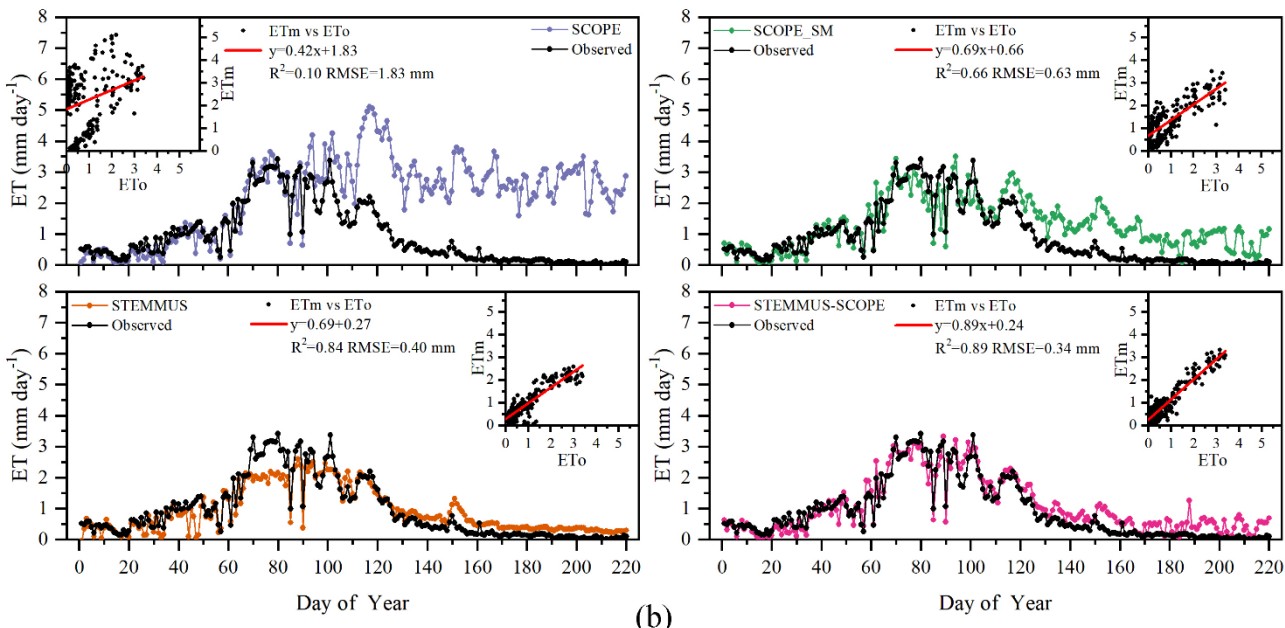

**Figure 6 Comparison of modeled and observed daily evapotranspiration (ET): (a) Maize cropland at Yangling station; (b) Grassland at Vaira Ranch (US-Var) Fluxnet site. (ETm: modeled ET; ETo: observed ET).**

The modeled and observed daily transpiration at maize cropland are presented in Figure 7 and the modeled transpiration at grassland is presented in Figure 8. For Yangling station, the $R^2$ value between simulated and observed transpiration was 0.82, 0.86, 0.79, and 0.86, and the *RMSE* was 0.60, 0.50, 0.67, and 0.50 mm day$^{-1}$, for SCOPE, SCOPE_SM, STEMMUS and STEMMUS-SCOPE, respectively. Because of ignoring the effect of water stress on transpiration, SCOPE failed to simulate transpiration accurately when the vegetation experienced water stress. As shown in the Figure 6(a), SCOPE overestimated

transpiration for the maize cropland at Yangling station from DOY 183 to DOY 202 during the water stress period. Compared with SCOPE, SCOPE_SM, STEMMUS and STEMMUS-SCOPE can capture the reduction of transpiration during the dry period. The performances of STEMMUS-SCOPE and SCOPE_SM were also better than that of STEMMUS. The possible reason is the more processed-based consideration of the radiative transfer and energy balance at leaf level in the coupled STEMMUS-SCOPE model (as also in SCOPE_SM) and the more accurate root water uptake (compared to that in

SCOPE_SM). Nevertheless, STEMMUS-SCOPE slightly underestimated transpiration when the plant was undergoing severe water stress and slightly overestimated it after the field was irrigated. This is mainly because the actual $V_{cmax}$ was not only influenced by drought but also related to leaf nitrogen content (Xu and Baldocchi, 2003), which was not considered in the maize cropland simulation. Although the measured T at the grassland was not available, we compared modeled T by the four models (Figure 7). During the wet season (before DOY 85), the modeled T by SCOPE, SCOPE_SM, and STEMMUS-SCOPE

were similar and were higher than that by STEMMUS from DOY 64 to 82. During the dry season (after DOY 85), due to the simplified consideration of soil processes, the modeled T by SCOPE and SCOPE_SM were both much higher than that by

STEMMUS and STEMMUS-SCOPE. The reason for the better performance by the coupled model for the grassland (Figure 6(b)) is that it considers also the effect of leaf chlorophyll content ($C_{ab}$) on $V_{cmax}$, in addition to more detailed consideration of water stress as discussed above for the maize cropland.

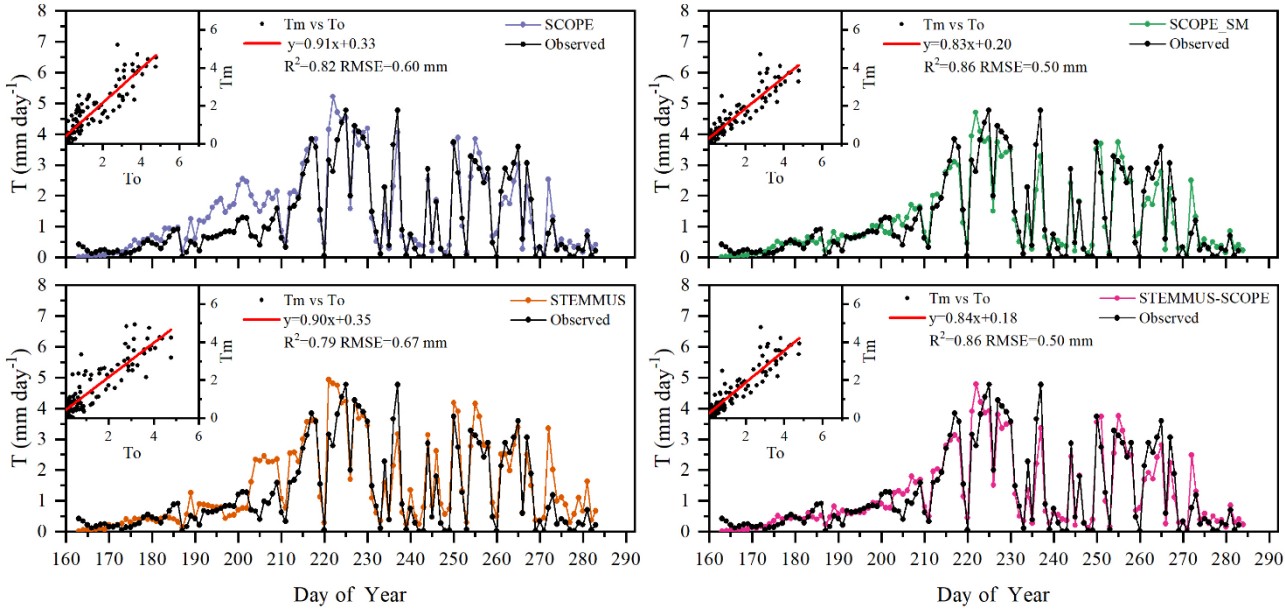

**Figure 7 Comparison of modeled and observed daily plant transpiration (T) for the maize cropland at Yangling station (Tm: modeled T; To: observed T).**

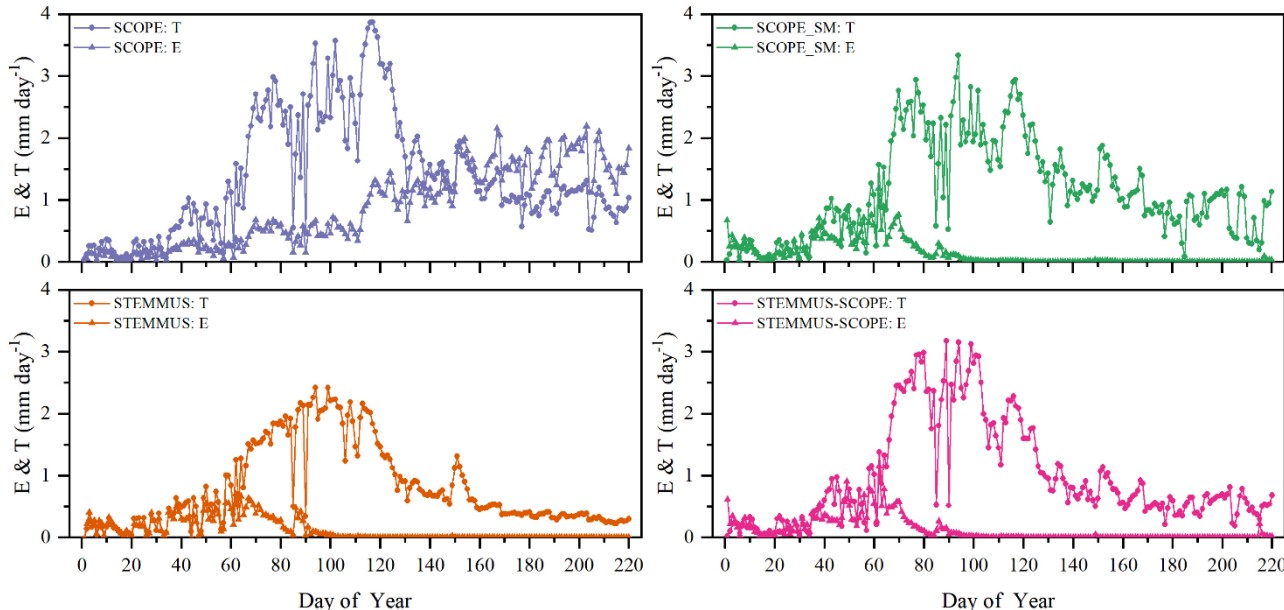

**Figure 8 Comparison of modeled daily transpiration (T) and soil evaporation (E) for grassland at Vaira Ranch (US-Var) Fluxnet site (T: transpiration; E: soil evaporation).**

As shown in Figure 9 for soil evaporation at Yangling station, the simulation by STEMMUS-SCOPE is closer to observation than those by other models. When using SCOPE to simulate soil evaporation, the soil moisture is set as constant (i.e., 0.25 m$^3$ m$^{-3}$). Therefore, SCOPE generally underestimates soil evaporation when soil moisture is higher than 0.25 and overestimates it when it is lower than 0.25. Here we use the average soil moisture at root zone simulated by STEMMUS-SCOPE as the input data for SCOPE and SCOPE_SM to calculate soil surface resistance and soil evaporation. Although STEMMUS can capture variation of soil evaporation reasonably well, it has higher *RMSE* value than STEMMUS-SCOPE. This is probably attributed to the comprehensive consideration of radiation transfer in SCOPE, which is lacking in STEMMUS. Consequently, the simulation of soil net radiation by the coupled model was more accurate than that by STEMMUS alone. The *RMSE* value by STEMMUS-SCOPE was 0.60 mm day$^{-1}$, which was lower than those by other three models (i.e. 0.67, 0.65, and 0.64 mm day$^{-1}$ respectively). For STEMMUS-SCOPE, the major differences between simulations and observations occurred in rainy or irrigation days (cf. Figure 2(a)), which may be caused by errors of the estimated soil surface resistance during these periods or the uncertainty of ET partitioning method. The uncertainty of ET partitioning method (Zhou et al., 2016) was mainly caused by: (1) the uncertainty in the partitioning of *GPP* (less than 10%) and *Re* based on *NEE*, which would result in some uncertainty in *uWUE*; (2) due to the seasonal variation of atmosphere $CO_2$ concentration, the assumption of *uWUE$_p$* being constant would cause some uncertainty (less than 3%); (3) the assumption of *T* being equal to *ET* sometimes during the growing season would cause some uncertainty when vegetation is sparse. Because the observed E at US-Var site was not available, a comparison of

only modeled E was shown in Figure 8, in which SCOPE modeled unrealistic E during the dry season, while the modeled E by SCOPE_SM, STEMMUS, and STEMMUS-SCOPE were consistent due to use the simulated surface *SM* as the input for soil evaporation calculation.

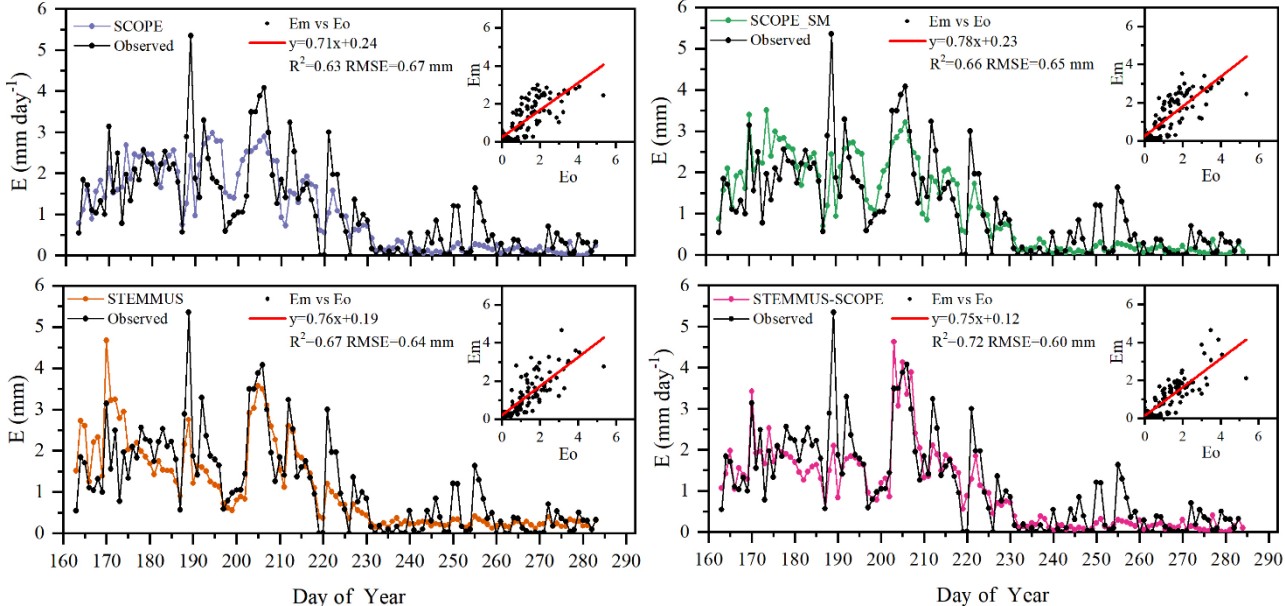

**Figure 9 Comparison of modeled and observed daily soil evaporation (E) at Yangling station (Em: modeled E; Eo: observed E).**

### 3.5. Daily GPP modelling

Simulated *GPP* by SCOPE, SCOPE_SM and STEMMUS-SCOPE and observed *GPP* are presented in Figure 10. As shown, similar to the simulation of transpiration, SCOPE cannot respond to water stress when simulating *GPP*. After introducing soil water stress factor in STEMMUS-SCOPE and SCOPE_SM, the simulations of *GPP* were improved in both models. For Yangling station, the consistency between simulated and observed *GPP* at mid and late stages were higher than those at early and rapidly growth stages. The difference usually occurred when soil moisture increased. For US-Var site, STEMMUS-SCOPE can simulate *GPP* well during the whole period, while SCOPE_SM slightly underestimated *GPP* around DOY 80 when this site transits from wet season to dry season. It indicates that only using the surface *SM* cannot reflect the actual root zone *SM* when the vegetation experiencing moderate water stress. Under such a condition, the hydraulic redistribution (HR) and compensatory root water uptake (CRWU) process enable the vegetation to utilize the water in deep soil layer. Only using the surface soil water content to calculate RWU in SCOPE_SM ignored the effect of HR and CRWU process, and the effect of water stress was overestimated. However, the surface soil moisture can reflect root zone soil moisture well when the vegetation was not under water stress or severe water stress. A similar underestimated of *GPP* was also found by Bayat et al. (2019).

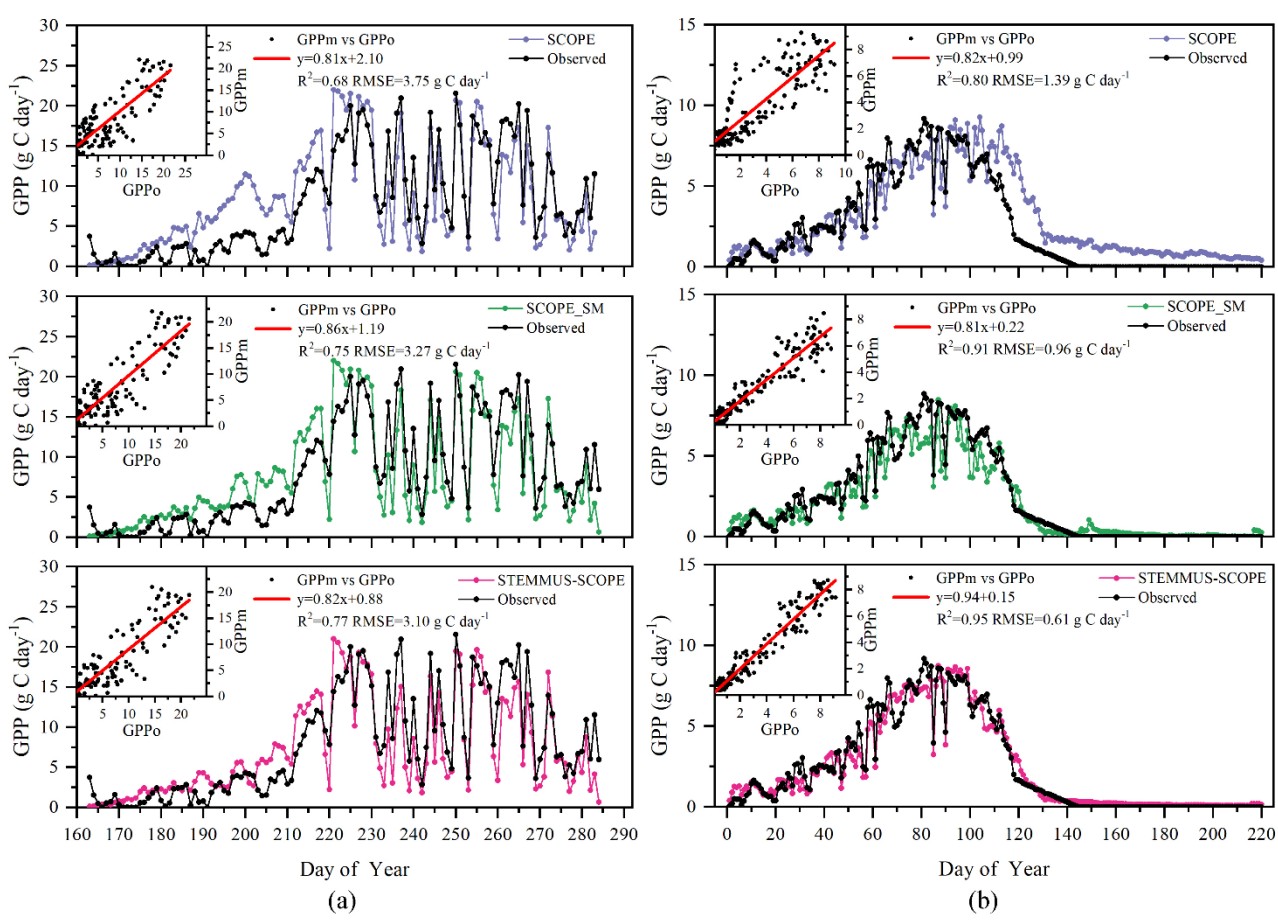

**Figure 10 Comparison of modeled and observed daily gross primary production (GPP): (a) Maize cropland at Yangling station; (b) Grassland at Vaira Ranch (US-Var) Fluxnet site. (GPPm: modeled GPP; GPPo: observed GPP).**

### 3.6. Simulation of leaf water potential (LWP), water stress factor (WSF), and root length density (RLD)

The simulated half-hourly leaf water potential and water stress factor at Yangling station are presented in Figure 11. The leaf water potential was lower when vegetation suffering water stress compared to other periods. The reason is that soil water potential is low due to the low soil moisture and the plants need to maintain an even lower leaf water potential to suck water from the soil and transfer it to leaves. During mid and late stages, the leaf water potential was sensitive to transpiration demand due to the slowdown of root system growth. As the continuous measurements of the leaf water potential is not available, we

compared only the magnitude of simulated leaf water potential to measurements reported in literature.

Many studies have measured midday leaf water potential or dawn leaf water potential. Fan et al. (2015) reported that the leaf water potential of well-watered maize was maintained high between -73 to -88 m and leaf water potential would decrease when soil water content was lower than 80% of field capacity. Martineau et al. (2017) reported the midday leaf water potential

of well-watered maize was around --0.82 MPa (about 84.8 m in water pressure head; note: 0.1MPa equal to 10.339 m water pressure head) and the midday leaf water potential decreased to --1.3 MPa (about 134.4 m in water head) when the maize was suffering water stress. Moreover, O'Toole and Cruz (1980) studied the response of leaf water potential to water stress in rice and concluded that the leaf water potential of rice can be lower than -80 to -120 m when the vegetation was under water stress and the leaves started curling, which was similar to the simulated leaf water potential of maize in this study. Aston and Lawlor (1979) revealed the relationship between transpiration, root water uptake and leaf water potential of maize. These field studies found that leaf water potential was often very low and reached trough values at midday. Elfving (1972) developed a water flux model based on SPAC system, evaluated it for orange tree, and reported about -120 m for the trough value of leaf water potential under non-limiting environmental conditions, which was slightly lower than the simulation in this study.

In this study, the calculation of water stress factor considered the effect of soil moisture and root distribution. The severe water stress occurred from DOY 183 to DOY 202, and the coupled model performed very well in this period. As the feedback, water stress can also influence root water uptake and root growth, and consequently influence soil moisture and root dynamics in next time step. It indicates that the water stress equation used in this study can characterize the reduction of $V_{cmax}$ reasonably well.

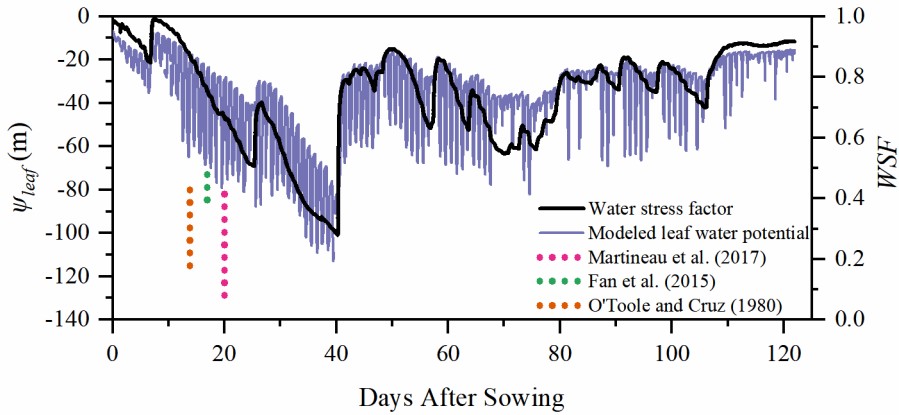

**Figure 11 Simulation of $\psi_{leaf}$ (leaf water potential, m) and $WSF$ (water stress factor) at Yangling station. (The dashed lines represent the range of midday leaf water potential reported in other sites.)**

Root length density is another vital parameter in calculating root water uptake. As shown in Table 3, the simulated peak root length density and maximum rooting depth of maize at Yangling station was comparable to the measured values at other sites. Many previous studies revealed that root length density was influenced by soil moisture, bulk density, tillage, and soil mineral nitrogen (Amato and Ritchie, 2002; Chassot et al., 2001; Schroder et al., 1996). In this study, as we assumed the soil was homogenous, STEMMUS-SCOPE considered the effect of soil moisture but neglected the effect of bulk density and soil mineral nitrogen. Amato and Ritchie (2002) also found a similar result as this study about the root length density in a maize field. Peng et al. (2012) studied temporal and spatial dynamics in root length density of field-grown maize and found that 80%

root length density was distributed at 0-30 cm depth with peak values from 0.86 to 1.00 cm cm$^{-3}$. Ning et al. (2015) also reported a similar observation of root length density. Chassot et al. (2001) and Qin et al. (2006) reported that root length density

can reach 1.59 cm cm$^{-3}$ at Swiss midlands. In Stuttgart, Germany, Wiesler and Horst (1994) observed the root growth and nitrate utilization of maize under field condition. The observed root length density was 2.45-2.80 cm cm$^{-3}$ at 0-30 cm depth which was much higher than in other studies, and decreased to 0.01 cm cm$^{-3}$ at 120-150 cm depth, which was consistent with the observation of Oikeh et al. (1999) at Samaru, Nigeria. Zhuang et al. (2001b) proposed a scaling model to estimate the distribution of root length density of field grown maize. In their study, measured root length density in Tokyo, Japan decreased

from 0.4-0.95 cm cm$^{-3}$ at top soil layer to about 0.1 cm cm$^{-3}$ at the bottom layer. Zhuang et al. (2001a) observed that the root length density of maize was mainly distributed at 0-60 cm depth and the maximum values were about 0.9 cm cm$^{-3}$. These studies indicated that the root length density values were quite variable when it was observed at different sites, nevertheless the simulated root length density in our study was in order of magnitude similar to the observations in previous studies (Table 3).

**Table 3 Comparison of the peak root length density (RLD) (cm cm$^{-3}$) at Yangling station with that at other sites.**

| Location | Maximum rooting depth (cm) | Peak *RLD* (cm cm$^{-3}$) | Soil type | Bulk density (g cm$^{-3}$) | References |
|---|---|---|---|---|---|
| Potenza, Italy | 100 | 0.84 | Clay-loam | 1.59-1.69 | Amato and Ritchie (2002) |
| Beijing, China | 60 | 0.78 | Silty loam | | Peng et al. (2012) |
| Alize, Stuttgart, Germany | 150 | 2.45 | Clay | 1.5-1.7 | Wiesler and Horst (1994) |
| Brummi, Stuttgart, Germany | 150 | 2.80 | Clay | 1.5-1.7 | Wiesler and Horst (1994) |
| Swiss midlands | 100 | 1.59 | Sandy silt | 1.21-1.55 | Qin et al. (2006) |
| Samaru, Nigeria | 90 | 2.78 | Loamy soil | 1.39-1.67 | Oikeh et al. (1999) |
| Tokyo, Japan | 58 | 0.95 | Sandy loam | 0.61-0.80 | Zhuang et al. (2001a, b) |
| Yangling, China | 121 | 0.74 | Sandy loam | 1.41 | This study |

### 3.7. Diurnal variation of T, GPP, SIF, and LWP

Figure 12 shows the modeled and observed half-hourly canopy transpiration (*T*), gross primary production (*GPP*), solar-induced fluorescence (*SIF*) and leaf water potential (*LWP*) from DOY 183 to 202 at Yangling station. The simulations by STEMMUS-SCOPE and SCOPE_SM were consistent with observation while that by SCOPE was much higher than

observation. The performances of STEMMUS-SCOPE and SCOPE_SM were consistent with that of SCOPE in the early morning and late afternoon, when the photosynthesis was mainly limited by incident radiation rather than by water stress, intercellular $CO_2$ concentration and $V_{cmax}$. In the midday, with increasing incident radiation, the photosynthesis was mainly limited by water stress and $V_{cmax}$, exactly when the simulations by STEMMUS-SCOPE and SCOPE_SM were much better than that by SCOPE. The diurnal variation of observed and modeled *GPP* were similar to that of T. Due to lack of observed

*SIF*, only the simulated *SIF* were presented. As the figure shown, the *SIF* simulated by STEMMUS- SCOPE and SCOPE_SM were reduced when the vegetation experiencing water stress, which indicated that both the simulated *SIF* of STEMMUS-SCOPE and SCOPE_SM can respond to water stress. However, the accuracy of the simulated *SIF* needs further validation with field observation.

Figure 13 shows the relationship among half-hourly *GPP*, *SIF*, and *LWP* on DOY 199 at Yangling station. There was a strong
linear relationship between *SIF* and *GPP* when the maize was well-watered (Figure 13a). However, *SIF* kept increasing while *GPP* tended to saturate when the maize suffering water stress. This result is consistent with the previous study conducted for cotton and tobacco leaves (Van der Tol et al., 2014). Because SCOPE_SM used the averaged root zone *SM* and ignored vertical root and soil water distribution, it overestimated *GPP* and *SIF*. When the maize was experiencing drought, the *LWP* was maintained at a low level. With *GPP* and T increasing, the plant decreased *LWP* in order to extract enough water from the root
zone. SPAC system enabled STEMMUS-SCOPE simulate half-hourly *LWP*. To better detect the response of simulated *SIF* to simulated *LWP*, we choose a cloudless day (DOY 199) and a liner relationship between the simulated *SIF* and *LWP* was obtained (Figure 13b). Sun et al. (2016) reported that *SIF*-soil moisture-drought relationship depended on variations of both absorbed *PAR* and fluorescence yield in response to water stress, while the *LWP* can reflect both effect of absorbed *PAR* and soil moisture status. The strong correlation between *GPP*, *LWP* and *SIF* indicates a potential of using *SIF* as an effective signal
for characterizing the response of photosynthesis to water stress. In the future, more studies should focus on the measurements of *SIF*, *GPP*, and *LWP* simultaneously for different vegetation types across different environmental conditions (radiation, soil moisture, and $CO_2$ concentration) to reveal how the water stress affects these relationships.

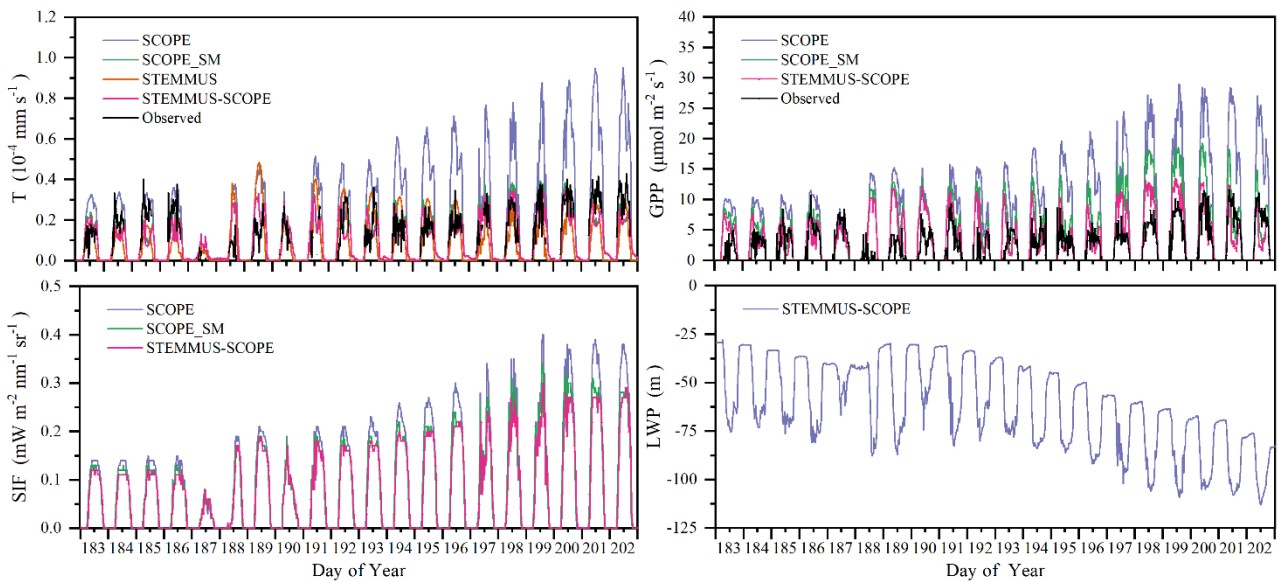

**Figure 12 Comparison of modeled and observed half-hourly transpiration (T), gross primary production (GPP), top of canopy solar-induced fluorescence (SIF) and leaf water potential (LWP) at Yangling station.**

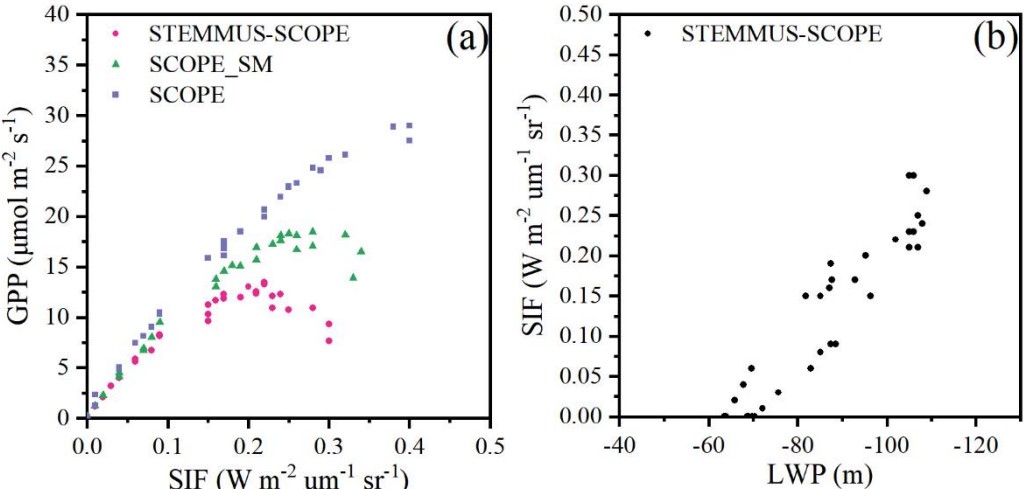

**Figure 13 The relationship among gross primary production (GPP), top of canopy solar induced fluorescence (SIF), and leaf water potential (LWP) on DOY 199: (a) GPP vs SIF; (b) SIF vs LWP.**

### 3.8. Limitations need to be overcame

The new coupled model notably improved simulations of carbon and water fluxes when vegetation suffering water stress. However, this study mainly aimed to improve the response of SCOPE to drought by introducing vertical soil water and root profile. Some critical processes were followed that existed in SCOPE_SM and STEMMUS. As with any model, some modules in STEMMUS-SCOPE, such as plant hydraulics and root growth, could be improved upon in future development.

First, to date many LSMs (e.g. CLM 5, Noah-MP, JULES, and CABLE) have incorporated state-of-the-art plant hydraulics
model to replace the conventional empirical plant hydraulic model which was only based on the distribution of *SM* and fraction of roots (e.g. CLM 4.5 and CoLM) (De Kauwe et al., 2015). Although STEMMUS-SCOPE integrated a 1D root growth model and a relatively novel RWU model, its hydraulics model followed that in SCOPE_SM and ignored the most exciting recent advances in our understanding of plant hydraulics: hydraulic failure due to loss of hydraulic conductivity due to embolism and refilling for recovery from xylem embolism (McDowell et al., 2019). Because STEMMUS-SCOPE performed well in maize
cropland and grassland, the influence of embolism and refilling on water transfer from the soil through vegetation to the atmosphere cannot be fully detected. The value of using plant water potential instead of soil water potential to constrain model predictions has been demonstrated in many case studies (De Kauwe et al., 2020; Niu et al., 2020; Medlyn et al., 2016; Xu et al., 2016; Williams et al., 1996). Niu et al. (2020) followed the plant hydraulic model developed by Xu et al. (2016) and represented plant stomatal water stress factor as a function of the plant water storage. CLM 5.0 also introduced a new
formulation for *WSF*, which is based on leaf water potential ($\psi_l$) instead of soil water potential ($\psi_s$) (Kennedy et al., 2019). These new formulations based on plant water potential could have significant improvements for plant drought responses. Besides, STEMMUS-SCOPE presently does not account for plant water storage that may result in underestimating morning *LE* and overestimating afternoon *LE*. Some field observations showed that the plant did not immediately respond when soil moisture was enhanced (Mackay et al., 2019), but there are long lags, which was ignored in this study too, between soil water
recovery from drought and plant responses to the recovery. The *WSF* in STEMMUS-SCOPE directly comes from soil moisture and cannot reflect true stomatal response when vegetation experiencing drought. For example, in early morning, the low stomatal aperture was induced by low *PAR* rather than by *SM*. Consequently, STEMMUS-SCOPE needs to introduce the advanced hydraulics when the model was tested in a wide range of ecosystems, particularly for vegetation exposed to frequent drought cycles or prolonged periods of severe drought events. It is important however to note that explicit representations of
plant hydraulics require additional model parameters and increase parameterization burden. This is the most challenging limitation to STEMMUS-SCOPE for incorporating these hydraulics models and we have chosen a trade-off between mechanism and practicality.

Second, as mentioned above, STEMMUS-SCOPE adapted the macroscopic RWU model and a simplified 1D root growth model for saving computational costs, though it well predicted maximum root depth which is the most critical factor when
calculating *WSF* and RWU. Such a simplification would likely ease the migration of our model into larger-scale models, such

as earth system models. However, STEMMUS-SCOPE oversimplified metabolic processes of the roots that include root exudates, root maintenance respiration, root growth respiration, and root turnover, which are also critical and have been incorporated in Noah-MP (Niu et al., 2020). This simplification could result in uncertainties in modelling the root growth and root water uptake. Meanwhile, there was no validation of seasonal vertical root length distribution based on in-situ observation, which need to be validated in the next step. Furthermore, the model presently does not account for the feedback between hydraulic controls over carbon allocation and the role of root growth on soil-plant hydraulics, which could also be considered in future model development.

## 4. Conclusions

A fundamental understanding of coupled energy, water and carbon flux is vital for obtaining the information of ecohydrological processes and functioning under climate change. The coupled model, STEMMUS-SCOPE, integrating radiative transfer, photochemistry, energy balance, root system dynamics, and soil moisture and soil temperature dynamics, has been proven to be a practical model to simulate detailed land surface processes such as evapotranspiration and *GPP*. In the coupled model, STEMMUS could provide root zone moisture profile to SCOPE, which was used to calculate water stress factor. On the other hand, SCOPE can provide net carbon assimilation and soil surface temperature to STEMMUS, which was used subsequently as the top boundary condition and as the input for root growth model. This study explores the role of dynamic root growth in affecting canopy photosynthesis activities, fluorescence emissions and evapotranspiration, which has not been reported before. The coupled model has been successfully applied in a maize field and a grassland, and can be used to describe ET partitioning, canopy photosynthesis, reflectance, and fluorescence emissions. The results show that via considering dynamic root growth and the associated root water uptake, the simulated *SIF* of the coupled STEMMUS-SCOPE model can response to water stress, while this is not the case for SCOPE_SM.

Through the inter-comparison of SCOPE, SCOPE_SM, STEMMUS, and STEMMUS-SCOPE, we concluded that the coupled STEMMUS-SCOPE can be used to investigate vegetation states under water stress conditions, and to simultaneously understand the dynamics of soil heat and mass transfer, as well as the root growth. By considering vertical distribution of soil moisture and root system, the simulation of water and carbon fluxes, especially when vegetation suffering moderate water stress, was significantly improved. However, there remain some needs for further studies to enhance the capacity of STEMMUS-SCOPE in understanding ecosystem functioning. Frist of all, the estimation of soil boundary condition especially during the irrigation period, which has significant influence on the simulation of soil temperature, needs further considerations. Second, the realism of the present model in modelling water-stressed *SIF* are subject to further studies. Nevertheless, STEMMUS-SCOPE may be used as an effective forward simulator to simulate remote sensing signals and to assimilate remote sensing data such as solar-induced chlorophyll fluorescence, to improve the estimation of water and carbon fluxes.

STEMMUS-SCOPE could also be used to investigate regional or global land surface processes, especially in arid and semi-arid regions, due to its sensitivity to water stress conditions.

*Code and data availability.* The development and validation of STEMMUS-SCOPE in this paper were conducted in MATLAB R2016a. The exact version of the model used to produce the results used in this paper is archived on Zenodo (Wang et al., 480 2020). The original source of the SCOPE model and STEMMUS model was obtained from Van der Tol et al. (2009) and Zeng et al. (2011a, b), respectively. The tower-based eddy-covariance measurements used for model validation were obtained from the authors for the Yangling station, China (Wang et al., 2019), from the FLUXNET2015 Dataset and PLUMBER2 program for the Vaira Ranch (US-Var) Fluxnet site.

*Author contributions.* YW, YZ, HC and ZS designed the study, YW developed the code, conducted the analysis, and wrote 485 the manuscript, YW and HC collected and shared their eddy-covariance measurements for the purpose of model validation. All authors discussed, commented and contributed to the revisions and final version of the manuscript.

*Competing interests.* The authors declare that they have no conflict of interest.

*Acknowledgments.* This work was supported by the National Natural Science Foundation of China (51879223 and 41971033), the National Key Research and Development Program of China (2016YFC0400201), the Fundamental Research Funds for the 490 Central Universities, CHD (300102298307), and China Scholarship Council. Peiqi Yang was supported by the Netherlands Organization for Scientific Research, grant ALWGO.2017.018.

**Appendix A**

**A.1. Photosynthesis and evapotranspiration under water stress in SCOPE**

The C4 Photosynthesis is calculated in the SCOPE model as the minimum of three processes (Collatz et al., 1991; 1992); (1) carboxylation rate limited by Ribulose biphosphate-carboxylase-oxygenase activity (known as Rubisco (enzyme)-limited, $V_c$, described in Eq. (A1); (2) carboxylation rate limited by Ribulose 1–5 bisphosphate regeneration rate (known as RuBP (electron transport/light)-limited), $V_e$, described in Eq. (A2); (3) at low $CO_2$ concentrations, carboxylation rate limited by intercellular $CO_2$ partial pressure ($p_i$), $V_s$, described in Eq. (A3).

$$V_c = V_{cmax} * WSF \tag{A1}$$

$$V_e = \frac{J}{6} \frac{-b \pm \sqrt{b^2 - 4ac}}{2a} \tag{A2}$$

$$V_s = p_i (k_p - \frac{L}{p_i})/P \tag{A3}$$

$$A_n = min(V_c, V_e, V_s) \tag{A4}$$

The C3 Photosynthesis is calculated in the SCOPE model as the minimum of two processes (Farquhar et al., 1980); (1) carboxylation rate limited by Ribulose biphosphate-carboxylase-oxygenase activity (known as Rubisco (enzyme)-limited, $V_c$, described in Eq. (A5); (2) carboxylation rate limited by Ribulose 1–5 bisphosphate regeneration rate (known as RuBP (electron transport/light)-limited), $V_e$, described in Eq. (A6).

$$V_c = V_{cmax} * WSF * \frac{C_i - \Gamma^*}{C_i + K_c(1 + \frac{O_i}{K_o})} \tag{A5}$$

$$V_e = \frac{J(C_i - \Gamma^*)}{4(C_i + 2\Gamma^*)} \frac{-b \pm \sqrt{b^2 - 4ac}}{2a} \tag{A6}$$

$$A_n = min(V_c, V_e) \tag{A7}$$

$$C_i = C_a(1 - \frac{1}{mRH}) \tag{A8}$$

where $V_{cmax}$ is the maximum carboxylation rate (µmol m$^{-2}$ s$^{-1}$), $p_i$ is the intercellular $CO_2$ partial pressure (Pa), $k_p$ is a pseudo-first-order rate constant for PEP carboxylase with respect to $C_i$, $P$ is the atmospheric pressure; $A_n$ is the net photosynthesis (µmol m$^{-2}$ s$^{-1}$); $WSF$ is the total water stress factor, $J$ is the electron transport rate (µmol m$^{-2}$ s$^{-1}$), $C_i$ is the intercellular $CO_2$ concentration (µmol m$^{-3}$) and $C_a$ is $CO_2$ concentration in the boundary layer (µmol m$^{-3}$), $m$ is Ball-Berry parameter and RH is relative humidity at the leaf surface (%).

In addition, leaf stomatal resistance $r_c$ (s m$^{-1}$) is calculated as:

$$r_c = \frac{0.625(C_s - C_i)}{A_n} \frac{\rho_a}{M_a} \frac{10^{12}}{p} \tag{A9}$$

Where $\rho_a$ is specific mass of air (kg m$^{-3}$), $M_a$ is molecular mass of dry air (g mol$^{-1}$), and $p$ is atmosphere pressure (hPa).

The calculation of latent heat flux (*LE*) is as follows:

$$LE = \lambda \frac{(q_i - q_a)}{r_a + r_c} \tag{A10}$$

Where $\lambda$ is vaporization heat of water (J kg$^{-1}$), $q_i$ is the humidity in stomata or soil pores (kg m$^{-3}$), $q_a$ is the humidity above the canopy (kg m$^{-3}$), $r_c$ is stomatal or soil surface resistance (s m$^{-1}$), $r_a$ is aerodynamic resistance (s m$^{-1}$).

In the study of Bayat et al. (2019), water stress factor was calculated based on the root zone soil moisture content neglecting the distribution of root length. In this study, water stress factor considered both root length distribution and water content in root zone. We use a sigmoid formulation rather than the piecewise function by Bayat et al. (2019). The calculations are as follows:

$$WSF = \sum_{i=1}^{n} RF(i) * WSF(i) \tag{A11}$$

$$WSF(i) = \frac{1}{1 + e^{-100 * \theta_{sat}\left(SM(i) - \frac{\theta_f + \theta_w}{2}\right)}} \tag{A12}$$

$\theta_w$ is the soil water content at wilting point; $\theta_f$ is the soil water content at field capacity; $\theta_{sat}$ is the saturated soil water content; *WSF(i)* is the water stress factor at each soil layer; *RF(i)* is the ratio of root length in soil layer *i* and its calculation can be found in the appendix A.4; *SM(i)* is the soil moisture at each soil layer.

### A.2. Governing Equations in STEMMUS

### A.2.1 Soil water conservation equation

$$\frac{\partial}{\partial t}(\rho_L \theta_L + \rho_V \theta_V) = -\frac{\partial}{\partial z}(q_{Lh} + q_{LT} + q_{La} + q_{vh} + q_{vT} + q_{va}) - S = \rho_L \frac{\partial}{\partial z}\left[K\left(\frac{\partial h}{\partial z} + 1\right) + D_{TD}\frac{\partial T_s}{\partial z} + \frac{K}{\gamma_w}\frac{\partial P_g}{\partial z}\right] + \frac{\partial}{\partial z}\left[D_{Vh}\frac{\partial h}{\partial z} + D_{VT}\frac{\partial T_s}{\partial z} + D_{Va}\frac{\partial P_g}{\partial z}\right] - S \tag{A13}$$

where $\rho_L$, $\rho_V$ (kg m$^{-3}$) are the density of liquid water, water vapor, respectively; $q_L$, $q_V$ (m$^3$ m$^{-3}$) are the volumetric water content (liquid and water vapor, respectively); z (m) is the vertical space coordinate (positive upwards); $S$ (cm s$^{-1}$) is the sink term for the root water extraction. $K$ (m s$^{-1}$) is hydraulic conductivity; $h$ (cm) is the pressure head; $T_s$ (°C) is the soil temperature; and $P_g$ (Pa) is the mixed pore-air pressure. $\gamma_w$ (kg m$^{-2}$ s$^{-2}$) is the specific weight of water. $D_{TD}$ (kg m$^{-1}$ s$^{-1}$ °C$^{-1}$) is the transport coefficient for adsorbed liquid flow due to temperature gradient; $D_{Vh}$ (kg m$^{-2}$ s$^{-1}$) is the isothermal vapor conductivity; and $D_{VT}$ (kg m$^{-1}$ s$^{-1}$ °C$^{-1}$) is the thermal vapor diffusion coefficient. $D_{Va}$ is the advective vapor transfer coefficient (Zeng et al. 2011a,b).

$q_{Lh}$, $q_{LT}$, and $q_{La}$ (kg m$^{-2}$ s$^{-1}$) are the liquid water fluxes driven by the gradient of matric potential, temperature, and air pressure, respectively. $q_{Vh}$, $q_{VT}$, and $q_{Va}$ (kg m$^{-2}$ s$^{-1}$) are the water vapor fluxes driven by the gradient of matric potential, temperature, and air pressure, respectively.

## A.2.2 Dry air conservation equation

$$\frac{\partial}{\partial t}\left[\varepsilon\rho_{da}(S_a + H_cS_L)\right] = \frac{\partial}{\partial z}\left[D_e\frac{\partial\rho_{da}}{\partial z} + \rho_{da}\frac{S_aK_g}{\mu_a}\frac{\partial P_g}{\partial z} - H_c\rho_{da}\frac{q_L}{\rho_L} + \left(\theta_aD_{Vg}\right)\frac{\partial\rho_{da}}{\partial z}\right] \tag{A14}$$

where $\varepsilon$ is the porosity; $\rho_{da}$ (kg m$^{-3}$) is the density of dry air; $S_a$ (=1-$S_L$) is the degree of air saturation in the soil; $S_L$ (=$\theta_L/\varepsilon$) is the degree of saturation in the soil; $H_c$ is Henry's constant; $D_e$ (m$^2$ s$^{-1}$) is the molecular diffusivity of water vapor in soil; $K_g$ (m$^2$) is the intrinsic air permeability; $m_a$ ( kg m$^{-2}$ s$^{-1}$) is the air viscosity; $q_L$ (kg m$^{-2}$ s$^{-1}$) is the liquid water flux; $\theta_a$ (=$\theta_V$) is the volumetric fraction of dry air in the soil; and $D_{Vg}$ (m$^2$ s$^{-1}$) is the gas phase longitudinal dispersion coefficient (Zeng et al., 2011a,b).

## A.2.3 Energy balance equation

$$\frac{\partial}{\partial t}\left[(\rho_s\theta_sC_s + \rho_L\theta_LC_L + \rho_V\theta_VC_V + \rho_{da}\theta_aC_a)(T_s - T_r) + \rho_V\theta_VL_0\right] - \rho_LW\frac{\partial\theta_L}{\partial t} = \frac{\partial}{\partial z}\left(\lambda_{eff}\frac{\partial T}{\partial z}\right) - \frac{\partial}{\partial z}\left[q_LC_L(T_s - T_r) + q_V(L_0 + C_V(T_s - T_r)) + q_aC_a(T_s - T_r)\right] - C_LS(T_s - T_r) \tag{A15}$$

where $C_s$, $C_L$, $C_V$, $C_a$ (J kg$^{-1}$ °C$^{-1}$) are the specific heat capacities of solids, liquid, water vapor, and dry air, respectively; $\rho_s$ (kg m$^{-3}$), $\rho_L$ (kg m$^{-3}$), $\rho_V$ (kg m$^{-3}$), and $\rho_{da}$ (kg m$^{-3}$) are the density of solids, liquid water, water vapor, and dry air, respectively; $\theta_s$ is the volumetric fraction of solids in the soil; $\theta_L$, $\theta_V$, and $\theta_a$ are the volumetric fraction of liquid water, water vapor, and dry air, respectively; $T_r$ (°C) is the reference temperature; $L_0$ (J kg$^{-1}$) is the latent heat of vaporization of water at temperature $T_r$; $W$ (J kg$^{-1}$) is the differential heat of wetting (the amount of heat released when a small amount of free water is added to the soil matrix); and $\lambda_{eff}$ (W m$^{-1}$ °C$^{-1}$) is the effective thermal conductivity of the soil; $q_L$, $q_V$, and $q_a$ (kg m$^{-2}$ s$^{-1}$) are the liquid, vapor water and dry air flux.

## A.3. Dynamic Root Growth Modelling

### A.3.1. Root front growth

The depth of the root front is firstly initialized either with the sowing depth for sown crops or with an initial value for transplanted crops or perennial crops. The root front growth stops when it reached certain depth of soil or a physical/chemical obstacle preventing root growth, but also stops when the phenological stopping stage has been reached.

$$\Delta Z = \begin{cases} 0 & T_{air} < T_{min} \\ (T_{air} - T_{min}) * RGR & T_{min} < T_{air} < T_{max} \\ (T_{max} - T_{min}) * RGR & T_{max} < T_{air} \end{cases} \tag{A16}$$

$$D_Z(t) = D_Z(t-1) + \Delta Z \tag{A17}$$

where $\Delta Z$ is root front growth at $t$-th time step; $D_Z$ (cm) is root zone depth; $T_{air}$ ($^0$C) is air temperature; $T_{min}$ ($^0$C) is the minimum temperature for root growth; $T_{max}$ ($^0$C) is the maximum temperature for root growth; $RGR$ (cm $^0$C$^{-1}$ day$^{-1}$) is the root growth rate of root front.

### A.3.2. Root length growth

In this study, the root distribution in the root zone was realized via simulating the root length growth in each soil layer.

$$\Delta Rl\_tot = \frac{A_n * fr_{root}}{R_C * R_D * \pi * r_{root}^2} \tag{A18}$$

$fr_{root}$ is the allocation fraction of net assimilation to root, and $fr_{root}$ is assumed as a function of leaf area index (LAI) and root zone water content. $A_n$ is the net assimilation rate ($\mu$mol m$^{-2}$ s$^{-1}$). $R_C$ is ratio of carbon to dry organic matter in root, $R_D$ is root density (g m$^{-3}$), and $r_{root}$ is radius of the root, and $\Delta Rl\_tot$ (m m$^{-3}$) is total root length growth.

The limiting factors for allocation are preliminarily computed and they account for root zone soil moisture availability $A_W$, and light availability $A_L$.

$$A_W = max[0.1, \quad min(1, \quad WSF)] \tag{A19}$$

where $WSF$ is the averaged soil moisture stress factor in the root zone.

$$A_L = max[0.1, \quad e^{-K_e \ LAI}] \tag{A20}$$

where $K_e = 0.15$ is a constant light extinction coefficient.

$$fr_{root} = max\left[r_{min}, \quad r_0 \frac{3A_L}{A_L + 2A_W}\right] \tag{A21}$$

where $r_{min}(= 0.15)$ is the minimum allocation coefficient to fine roots, and $r_0$ is a coefficient that indicates the theoretically unstressed allocation to fine roots.

$$\Delta Rl(i) = \Delta Rl\_tot * RF(i) \tag{A22}$$

where $RF(i)$ is the allocation fraction of root growth length in layer $i$, $\Delta Rl(i)$ is the root growth length in layer $i$.

For $i = 1$ to $n$-$1$ ($i = 1$ means the top soil layer):

$$Rl_i^t = Rl_i^{t-1} + \Delta Rl(i) \tag{A23}$$

For $i = n$:

$$Rl_i^t = Rl_i^{t-1} + \Delta Rl(i) + Rl_{front} \tag{A24}$$

where $Rl_i^t$ and $Rl_i^{t-1}$ is the root length of layer $i$ at time step $t$ and time step $t$-$1$.

$$RF(i) = \frac{Rl(i)}{Rl_T} \tag{A25}$$

where $Rl_T$ is the total root length in root zone, $Rl(i)$ is the root length in soil layer $i$.

At the root front, the density is imposed and estimated by the parameter $L_{v\_front}$ and the growth in root length depends directly on the root front growth rate $\Delta Z$:

$$Rl_{front} = L_{v\_front} * \Delta Z \tag{A26}$$

**A.4. Root water uptake**

The equation to calculate root water uptake and transpiration was as follows:

$$\sum_{i=1}^n \frac{\psi_{s,i} - \psi_l}{r_{s,i} + r_{r,i} + r_{x,i}} = \frac{0.622}{P} \frac{\rho_{da}}{\rho_V} \left(\frac{e_l - e_a}{r_c + r_a}\right) = T \tag{A27}$$

where $\psi_{s,i}$ is soil water potential of layer $i$ (pressure head, unit: m), $\psi_l$ is leaf water potential (m), $r_{s,i}$ is the soil hydraulic resistance (s m$^{-1}$), $r_{r,i}$ is the root resistance to water flow radially across the roots (s m$^{-1}$), and $r_{x,i}$ is the plant axial resistance to flow from the soil to the leaves (s m$^{-1}$). $e_l$ and $e_a$ are vapor pressure of leaf and the atmosphere (hPa), respectively, and $r_a$ and $r_c$ are aerodynamic resistance and canopy resistance (s m$^{-1}$), respectively. $\rho_{da}$ is the density of dry air (kg m$^{-3}$). $\rho_V$ is the density of water vapor. $P$ is the atmospheric pressure (Pa). 0.622 is the ratio of the molar mass of water to air.

$\psi_{s,i}$ is described as a function of soil moisture by Van Genuchten (1980), and the relevant parameters were shown in Table B.1.

The $r_s$ is calculated by Reid and Huck (1990) as:

$$r_s = \frac{1}{B \cdot K \cdot L_v \cdot \Delta d} \tag{A28}$$

where $B$ is the root length activity factor, $K$ is hydraulic conductivity of soil (m s$^{-1}$), $L_v$ is root length density (m m$^{-3}$), and $\Delta d$ is the thickness of the soil layer (m). $B$ is calculated as:

$$B = \frac{2\pi}{ln\left[(\pi R_D)^{-1/2}/r_{root}\right]} \tag{A29}$$

where $r_{root}$ is root radius (m).

The $r_r$ is estimated as (Reid and Huck, 1990):

$$r_r = \frac{P_r(\theta_{sat}/\theta)}{L_v \Delta d} \tag{A30}$$

where $P_r$ is root radial resistivity (s m$^{-1}$).

The xylem resistance $r_x$ is estimated by Klepper et al. (1983):

$$r_x = \frac{P_a Z_{mid}}{0.5 f L_v} \tag{A31}$$

where $P_a$ is root axial resistivity (s m$^{-3}$), $Z_{mid}$ is the depth of the midpoint of soil layer, and $f$ is a fraction defined for a specific depth as the number of roots which connect directly to the stem base to total roots crossing a horizontal plane at that depth. We can consider it equal to 0.22 based on Klepper et al. (1983).

The updated root water uptake term is:

$$S_i = \frac{\psi_{s,i} - \psi_l}{r_{s,i} + r_{r,i} + r_{x,i}} \tag{A32}$$

Different from other studies which need to calculate the compensatory water uptake and hydraulic redistribution after calculating the standard water uptake of each soil layer, the sink term in this study is calculated by a physically-based model which contain the effect of root resistance and soil hydraulic resistance rather than only considering the root fraction, so the compensatory water uptake and hydraulic redistribution have been considered when calculating the sink term.

**Appendix B.**

Table B.1 List of parameters and values used in this study (All the parameters were classified as Air, Canopy, Root and Soil).

| Symbol | Description | Unit | Value | |
|--------|-------------|------|-------|---|
| **Aerodynamic** | | | Maize | Grass |
| $aPAR$ | Absorbed photosynthetically active radiation | $\mu mol\ m^{-2}\ s^{-1}$ | | |
| $e_a$ | Air vapor pressure | Pa | | |
| $e_l$ | Vapor pressure of leaf | hPa | | |
| $P$ | Air pressure | Pa | | |
| $q_a$ | Humidity above the canopy | $kg\ m^{-3}$ | | |
| $q_l$ | Humidity in stomata | $kg\ m^{-3}$ | | |
| $r_a$ | Aerodynamic resistance | $s\ m^{-1}$ | | |
| $RH$ | Relative humidity | % | | |
| $R_{li}$ | Incoming longwave radiation | $W\ m^{-2}$ | | |
| $R_{in}$ | Incoming shortwave radiation | $W\ m^{-2}$ | | |
| $R_n$ | Net radiation | $W\ m^{-2}$ | | |
| $T_{air}$ | Air temperature | $^0C$ | | |
| $u$ | Wind speed | $m\ s^{-1}$ | | |
| $VPD$ | Vapor pressure deficit | hPa | | |
| **Canopy** | | | | |
| $A_n$ | Net assimilation rate | $\mu mol\ m^{-2}\ s^{-1}$ | | |
| $C_a$ | $CO_2$ concentration in the boundary layer | $\mu mol\ m^{-3}$ | | |
| $C_{ab}$ | Leaf chlorophyll content | $\mu g\ cm^{-2}$ | 80 | 0.374-50.45 |
| $C_{ca}$ | Leaf Carotenoid content | $\mu g\ cm^{-2}$ | 20 | $0.25*C_{ab}$ |
| $C_w$ | Leaf water content | $g\ cm^{-2}$ | 0.009 | .0.02 |
| $C_{dm}$ | Leaf dry matter content | $g\ cm^{-2}$ | 0.012 | 0.015 |
| $C_s$ | Senescent material content | | 0 | 0 |
| $DOY$ | Day of Year | d | | |
| $ET$ | Evapotranspiration | $mm\ day^{-1}$ | | |
| $GPP$ | Gross primary production | $g\ C\ m^{-2}\ day^{-1}$ | | |
| $h_c$ | Canopy height | m | 0-1.95 | 0.55 |
| $H$ | Sensible heat flux | $W\ m^{-2}$ | | |
| $J$ | Electron transport rate | $\mu mol\ m^{-2}\ s^{-1}$ | | |

| | | | | |
|---|---|---|---|---|
| $K_e$ | Light extinction coefficient | | 0.15 | 0.15 |
| $k_p$ | A pseudo-first-order rate constant for PEP carboxylase | | | |
| LAI | Leaf area index | $m^2\ m^{-2}$ | 0-4.39 | 0.745-2.03 |
| $LIDF$ | Leaf inclination distribution function | | -1, 0 | 0.08, -0.15 |
| $LE$ | Latent heat flux | $W\ m^{-2}$ | | |
| $LE_c$ | Latent heat flux of canopy | $W\ m^{-2}$ | | |
| $m$ | Ball-Berry stomatal conductance parameter | | 4 | 10 |
| $NEE$ | Net ecosystem exchange | $g\ C\ m^{-2}\ day^{-1}$ | | |
| $p_i$ | Intercellular $CO_2$ partial pressure | Pa | | |
| $r_c$ | Canopy resistance | $s\ m^{-1}$ | | |
| $Re$ | Ecosystem respiration | $g\ C\ m^{-2}\ day^{-1}$ | | |
| $T$ | Transpiration | $mm\ day^{-1}$ | | |
| $T_c$ | Vegetation temperature | $^0C$ | | |
| $T_{ch}$ | Leaf temperature (shaded leaves) | $^0C$ | | |
| $T_{cu}$ | Leaf temperature (sunlit leaves) | $^0C$ | | |
| $uWUE_p$ | Potential water use efficiency | $g\ C\ hPa^{0.5}/kg\ H_2O$ | | |
| $uWUE$ | Water use efficiency | $g\ C\ hPa^{0.5}/kg\ H_2O$ | | |
| $V_{cmax}$ | Maximum carboxylation rate | $\mu mol\ m^{-2}\ s^{-1}$ | 50 | 10.7-100.3 |
| $\psi_l$ | Leaf water potential | m | | |
| **Root** | | | | |
| $A_W$ | Root zone soil moisture availability | | | |
| $A_L$ | Light availability | | | |
| $B$ | Root length activity factor | | | |
| $D_Z$ | Root zone depth | cm | | |
| $f$ | A fraction defined for a specific depth as the number of roots which connect directly to the stem base to total roots crossing a horizontal plane at that depth | | 0.22 | 0.22 |
| $fr_{root}$ | Allocation fraction of net assimilation to root | | | |
| $P_a$ | Root axial resistivity | $s\ m^{-3}$ | $0.65*10^{12}$ | $2*10^{12}$ |
| $P_r$ | Root radial resistivity | $s\ m^{-1}$ | $1*10^{10}$ | $1.2*10^{11}$ |
| $RF(i)$ | The allocation fraction of root growth length in layer $i$ | | | |

| | | | | |
|---|---|---|---|---|
| $Rl_T$ | Total root length in root zone | m m$^{-2}$ | | |
| $Rl_i^t$ | Root length of layer $i$ at time step $t$ | m m$^{-2}$ | | |
| $Rl_i^{t-1}$ | Root length of layer $i$ at time step $t-1$ | m m$^{-2}$ | | |
| $Rl(i)$ | Root length in soil layer $i$ | m m$^{-2}$ | | |
| $Rl_{front}$ | Growth at the root front | m m$^{-2}$ | | |
| $RGR$ | Root growth rate of front | cm $^0$C day$^{-1}$ | 0.096 | 0.072 |
| $R_D$ | Root density | g m$^{-3}$ | 250000 | 250000 |
| $L_v$ | Root length density | m m$^{-3}$ | | |
| $L_{v\_front}$ | Root density at the root front | m m$^{-3}$ | 1000 | 150 |
| $r_{min}$ | The minimum allocation coefficient to fine roots | | 0.15 | 0.15 |
| $r_0$ | Coefficient of theoretically unstressed allocation to fine roots | | 0.3 | 0.3 |
| $r_{root}$ | Radius of the root | m | 0.15*10$^{-3}$ | 1.5*10$^{-3}$ |
| $r_{x,i}$ | Plant axial resistance to flow from the soil to the leaves | s | | |
| $r_{r,i}$ | Resistance to water flow radially across the roots | s | | |
| $r_{s,i}$ | Soil hydraulic resistance | s | | |
| $R_C$ | Ratio of carbon to dry organic matter in root | kg kg$^{-1}$ | 0.488 | 0.488 |
| RWU | Root water uptake | m s$^{-1}$ | | |
| $RF(i)$ | The ratio of root length in soil layer $i$ | | | |
| $T_{min}$ | Minimum temperature of root growth | $^0$C | 10 | 0 |
| $T_{max}$ | Maximum temperature of root growth | $^0$C | 40 | 40 |
| $\triangle Z$ | Root front growth at $t$-th step | cm | | |
| $\triangle Rl\_tot$ | Total root length growth | m | | |
| $\triangle Rl(i)$ | The root growth length in layer $i$ | m | | |
| **Soil** | | | | |
| $C_s$ | Specific heat capacities of solids | J kg$^{-1}$ $^{\circ}$C$^{-1}$ | | |
| $C_L$ | Specific heat capacities of liquid | J kg$^{-1}$ $^{\circ}$C$^{-1}$ | 4.186*10$^3$ | 4.186*10$^3$ |
| $C_V$ | Specific heat capacities of water vapor | J kg$^{-1}$ $^{\circ}$C$^{-1}$ | 1.870*10$^3$ | 1.870*10$^3$ |
| $C_a$ | Specific heat capacities of dry air | J kg$^{-1}$ $^{\circ}$C$^{-1}$ | 1.255*10$^{-3}$ | 1.255*10$^{-3}$ |
| $D_{TD}$ | Transport coefficient for absorbed liquid flow due to temperature gradient | kg m$^{-1}$ s$^{-1}$ $^{\circ}$C$^{-1}$ | | |
| $D_{Vh}$ | Isothermal vapor conductivity | kg m$^{-2}$ s$^{-1}$ | | |

| Symbol | Description | Units | | |
|---|---|---|---|---|
| $D_{VT}$ | Thermal vapor diffusion coefficient | kg m$^{-1}$ s$^{-1}$ °C$^{-1}$ | | |
| $D_{Va}$ | Advective vapor transfer coefficient | kg m$^{-2}$ s$^{-1}$ | | |
| $D_{Vg}$ | Gas phase longitudinal dispersion coefficient | m$^2$ s$^{-1}$ | | |
| $D_e$ | Molecular diffusivity of water vapor in soil | m$^2$ s$^{-1}$ | | |
| $E$ | Soil evaporation | mm | | |
| $G$ | Soil heat flux | W m$^{-2}$ | | |
| $h$ | Soil matric potential | cm | | |
| $H_c$ | Henry's constant | | 0.02 | 0.02 |
| $K$ | Hydraulic conductivity | m s$^{-1}$ | | |
| $K_g$ | Intrinsic air permeability | m$^2$ | | |
| $K_s$ | Saturation hydraulic conductivity | cm day$^{-1}$ | 18 | 10 |
| $LE_s$ | Latent heat flux of soil | W m$^{-2}$ | | |
| $L_0$ | Latent heat of vaporization of water temperature $T_r$ | J kg$^{-1}$ | 2497909 | 2497909 |
| $m_a$ | Air viscosity | kg m$^{-1}$ s$^{-1}$ | $1.846*10^{-5}$ | $1.846*10^{-5}$ |
| $n$ | Soil-dependent parameter | | 1.41 | 1.50 |
| $P_g$ | Mixed pore-air pressure | Pa | | |
| $q_L$ | Liquid water flux | kg m$^{-2}$ s$^{-1}$ | | |
| $q_{Lh}$ | Liquid water flux driven by the gradient of matric potential | kg m$^{-2}$ s$^{-1}$ | | |
| $q_{LT}$ | Liquid water flux driven by the gradient of temperature | kg m$^{-2}$ s$^{-1}$ | | |
| $q_{La}$ | Liquid water flux driven by the gradient of air pressure | kg m$^{-2}$ s$^{-1}$ | | |
| $q_V$ | Water vapor flux | kg m$^{-2}$ s$^{-1}$ | | |
| $q_{Vh}$ | Water vapor flux driven by the gradient of matric potential | kg m$^{-2}$ s$^{-1}$ | | |
| $q_{VT}$ | Water vapor flux driven by the gradient of temperature | kg m$^{-2}$ s$^{-1}$ | | |
| $q_{Va}$ | Water vapor flux driven by the gradient of air pressure | kg m$^{-2}$ s$^{-1}$ | | |
| $q_a$ | Dry air flux | kg m$^{-2}$ s$^{-1}$ | | |
| $S$ | Sink term for the root water extraction | cm s$^{-1}$ | | |
| $S_a$ | Degree of air saturation in the soil | | | |
| $S_L$ | Degree of saturation in the soil | | | |
| $SM(i)$ | The soil moisture at a specific soil layer | m$^3$ m$^{-3}$ | | |
| $T_s$ | Soil temperature | $^0$C | | |
| $T_{s0}$ | Soil surface temperature | $^0$C | | |

| | | | | |
|---|---|---|---|---|
| $T_r$ | Reference temperature | °C | 20 | 20 |
| $W$ | Differential heat of wetting | J kg$^{-1}$ | $1.001*10^3$ | $1.001*10^3$ |
| $WSF$ | Total water stress factor | | | |
| $WSF(i)$ | Water stress factor at a specific soil layer | | | |
| $Z_{mid}$ | The depth of the midpoint of soil layer | m | | |
| $\triangle d$ | Thickness of the soil layer | m | | |
| $\alpha$ | Soil-dependent parameter | m$^{-1}$ | 0.45 | 0.166 |
| $\theta_{sat}$ | Saturated water content | m$^3$ m$^{-3}$ | 0.42 | 0.38 |
| $\theta_f$ | Field capacity | m$^3$ m$^{-3}$ | 0.272 | 0.24 |
| $\theta_w$ | Wilting point | m$^3$ m$^{-3}$ | 0.10 | 0.03 |
| $\theta_r$ | Residual water content | m$^3$ m$^{-3}$ | 0.0875 | 0.0008 |
| $\theta$ | Volumetric soil water content | m$^3$ m$^{-3}$ | | |
| $\theta_L$ | Volumetric moisture content | m$^3$ m$^{-3}$ | | |
| $\theta_V$ | Volumetric vapor content | m$^3$ m$^{-3}$ | | |
| $\theta_s$ | Volumetric fraction of solids in the soil | m$^3$ m$^{-3}$ | | |
| $\theta_a$ | Volumetric fraction of dry air in the soil | m$^3$ m$^{-3}$ | | |
| $\psi_{s,i}$ | Soil water potential of layer $i$ | m | | |
| $\psi_{soil}$ | Soil water potential | m | | |
| $\lambda_{eff}$ | Effective thermal conductivity of the soil | W m$^{-1}$ °C$^{-1}$ | | |
| $\gamma_w$ | Specific weight of water | kg m$^{-2}$ s$^{-2}$ | | |
| $\rho_{da}$ | Density of dry air | kg m$^{-3}$ | | |
| $\rho_V$ | Density of vapor | kg m$^{-3}$ | | |
| $\rho_L$ | Density of liquid water | kg m$^{-3}$ | 1 | 1 |
| $\rho_s$ | Density of solids | kg m$^{-3}$ | | |
| $\varepsilon$ | Soil porosity | m$^3$ m$^{-3}$ | 0.50 | 0.50 |

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
