# Peer review of "Integrated Modeling of Photosynthesis and Transfer of Energy, Mass and Momentum in the Soil-Plant-Atmosphere Continuum System"

_Geoscientific Model Development, 2020_

## Short Comment (SC1) · 22 Jun 2020

Dear authors,

in my role as Executive editor of GMD, I would like to bring to your attention our Editorial version 1.2:

https://www.geosci-model-dev.net/12/2215/2019/

This highlights some requirements of papers published in GMD, which is also available on the GMD website in the 'Manuscript Types' section:

http://www.geoscientific-model-development.net/submission/manuscript_types.html

[Figure]

In particular, please note that for your paper, the following requirements have not been met in the Discussions paper:

- "The main paper must give the model name and version number (or other unique identifier) in the title."

- "If the model development relates to a single model then the model name and the version number must be included in the title of the paper. If the main intention of an article is to make a general (i.e. model independent) statement about the usefulness of a new development, but the usefulness is shown with the help of one specific model, the model name and version number must be stated in the title. The title could have a form such as, "Title outlining amazing generic advance: a case study with Model XXX (version Y)"."

As SCOPE is the model to which all your modifications and evaluation applies, SCOPE and a unique identifier, for the resulting version after integration of your modifications, should be named in the title of your article in your revised submission to GMD.

Yours,

Astrid Kerkweg

———————————————————

---

## Referee Comment (RC1) · Anonymous Referee #1 · 30 Jun 2020

The authors integrated photosynthesis and transfer of energy, mass and momentum in the soil-plant-atmosphere continuum system. The topic of this work is important, which may provide better insight into root water uptake modeling and the impacts of water availability on vegetation growth in terrestrial biosphere models. However, the manuscript still needs to be substantially improved.

1. SCOPE is a vertical (1-D) integrated radiative transfer and energy balance model, which is widely used in the simulations of vegetation photosynthesis process and fluorescence at the leaf and canopy level. The soil model is very simplified in SCOPE. It is interesting to see the STEMMUS Model, which is good at dealing with the mass and

heat transfer processes in unsaturated soil, is implemented into SCOPE. However, I do not see significant improvements in the SCOPE_STEMMUS model in current manuscript, although the SCOPE_STEMMUS includes root water uptake in unsaturated soil. I think this shortcoming lies in the model validations based on the measurements at the Yangling station. As we know, SCOPE model has the abilities to simulate the vegetation photosynthesis and evapotranspiration under the unstressed water conditions (Zhang et al., 2020; Shan et al., 2019; Zhang et al., 2018). And the Yangling station has irrigation and vegetation growth do not have water and heat stresses in 2017. Therefore, we can see very similar simulations from SCOPE and SCOPE_STEMMUS in Figure 6 and 7, which means the similar ability of SCOPE_STEMMUS and SCOPE in simulating ET and T. I think more sites that have water or heat stresses should be used for the validations to prove the better ability of SCOPE_STEMMUS.

2. For the development of SCOPE model, Bayat et al. (2019) have extended the SCOPE model to combine optical reflectance and soil moisture observations for remote sensing of ecosystem functioning under water stress conditions. Bayat's work has overcome the shortcoming in biased estimations of GPP and ET under water stressed conditions and the significant improvements of GPP and ET in SCOPE_SM model have shown in the paper. We also see the same abilities of SCOPE_SM and SCOPE_STEMMUS in simulating ET in this manuscript. Therefore, the authors should declare what the improvements are in SCOPE_STEMMUS model. Terrestrial biosphere models typically use empirical functions to represent vegetation responses to soil drought, especially in the water-limited areas. These functions largely neglect recent advances in plant ecophysiology that link xylem hydraulic functioning with stomatal responses to climate. I think this may be a direction to declare the new insights in the impacts of water tress on the vegetation growth, due to the descriptions of root water uptake in STEMMUS model.

3. In Table 1, some information should be updated. Nowadays, CLM5.0, CALBLE and JULES have large improvements in the hydraulic functioning with stomatal responses

to the warming climate (De Kauwe et al., 2020; Lawrence et al., 2020; Eller et al., 2020). And the authors should have more discussion about the root water uptake and the hydraulic functioning in the SCOPE_STEMMUS model in this manuscript.

4. The quality of some figures should be improved. This paper focus on the model developments and the better ability of the new model should be clear to the readers. For example, Figure 2 should be removed to the supplemental material. And Figure 5 and 8 are difficult for the readers to see and these figures should be redraw.

Reference

1. Zhang Z., Y.G. Zhang, A. Porcar-Castell, J. Joiner, L. Guanter, X. Yang, M. Migliavacca, et al. Reduction of structural impacts and distinction of photosynthetic pathways in a global estimation of GPP from space-borne solar-induced chlorophyll fluorescence. Remote Sensing of Environment, 2020, 111722.

2. Shan N., W. Ju, M. Migliavacca, D. Martini, L. Guanter, J.M. Chen, Y. Goulas, Y. Zhang. Modeling canopy conductance and transpiration from solar-induced chlorophyll fluorescence. Agricultural and Forest Meteorology, 2019, 268, 189-201.

3. Zhang Y., L. Guanter, J. Joiner, L. Song, K. Guan. Spatially-explicit monitoring of crop photosynthetic capacity through space-based chlorophyll fluorescence data. Remote Sensing of Environment, 2018,210, 362-374.

4. De Kauwe, M.G., Medlyn, B.E., Ukkola, A.M., Mu, M., Sabot, M.E., Pitman, A.J., Meir, P., Cernusak, L., Rifai, S.W., Choat, B., Tissue, D.T., Blackman, C.J., Li, X., Roderick, M. and Briggs, P.R. (2020), Identifying areas at risk of drought‐induced tree mortality across South‐Eastern Australia. Glob Change Biol.

5. Lawrence et al. (2020), Technical Description of version 5.0 of the Community Land Model (CLM).

6. Eller, Cleiton, Rowland, L., Mencuccini, Maurizio, Rosas, Teresa, Williams, Karina, Harper, Anna, Medlyn, Belinda, Wagner, Yael, Klein, Tamir, Teodoro, Grazielle,

Oliveira, Rafael, Matos, Ilaine, Rosado, Bruno H. P., Fuchs, Kathrin, Wohlfahrt, Georg, Montagnani, Leonardo, Meir, Patrick, Sitch, Stephen, Cox, Peter. (2020). Stomatal optimisation based on xylem hydraulics (SOX) improves land surface model simulation of vegetation responses to climate. New Phytologist. 226. 10.1111/nph.16419.

---

## Referee Comment (RC2) · Anonymous Referee #2 · 6 Jul 2020

It was difficult to see what are overarching scientific question and findings (including development of a novel model) in the current manuscript. Although the authors mentioned "most of the current vegetation photosynthesis models do not account for root water uptake, which compromises their applications under water, stressed conditions (P1L15-)", it should be noted that there are numerous SPAC models that are successful in taking into consideration the root water uptake (the authors should look at the pioneer paper (Williams et al. 1996, PCE 19, 911-927)). I think all figures shown in this manuscript can be reproduced by most existing SPAC models including most DGVMs, and thus, I feel they are meaningless to be represented.

[Figure]

Frankly speaking, because of the above reason I feel the current manuscript cannot be reviewed anymore, but I also feel this work is very potential. I acknowledge SCOPE has a huge advantage in terms of calculation of leaf-scale chlorophyll fluorescence. Thus, as the authors mentioned at the end of the manuscript (P21L392-), SCOPE_STEMMUS can be very state-of-the-arts SPAC model that can simulate the effect of plant water stress via soil moisture status on leaf-to-canopy scale chlorophyll fluorescence.

Thus, I will reject the current manuscript temporarily, but I strongly encourage the authors to resubmit this work with adding modelling results and discussion about the effect of plant water stress via soil moisture status on leaf-to-canopy scale chlorophyll fluorescence, which might be easily simulated using SCOPE_STEMMUS. For this, the authors should note: Obviously SCOPE_STEMMUS failed to reproduce the root developments (Fig. 12), but is successful in reproduction of transpiration and NEE. This is a serious inconsistency that prevents sound simulations of the effect of water stress on leaf gas exchange, and must be solved for resubmitting this work.

Though this is trivial point compared to the above-mentioned, I assumed the first author is an inexperienced scientist. For example, there was an ambiguous definition between "Results" and "Discussion" sections and were many wrong wordings. So I recommend to resubmit your paper to academic journals after thorough checking by the other experienced authors.

---

## Editor Comment (EC1) · Hisashi Sato (Editor) · 30 Jul 2020

As the topical editor of this manuscript, I do not encourage authors to submit the revision of this manuscript. Instead, I do recommend authors to decline this submission, and I encourage authors to resubmit it to the GMD if authors substantially improve the scientific significance of this study.

This recommendation is based on the reviewer's evaluation of the scientific significance of the manuscript and the negative comments in their review report. Normally, formal editorial recommendations or decisions are made after the authors have responded to all comments, or if they ask for editorial advice before responding. However, I strongly

believed that the second referee would be unlikely to change the evaluation and would further delay the manuscript.

Thank you for submitting your manuscript to GMD, and I hope you will consider submitting future work to GMD.

Sincerely yours, Hisashi SATO, Topical editor of the GMD

---

## Author Comment (AC1) · 23 Nov 2020

**Response to reviewer #1**

Thank you for the comprehensive comments, and also for taking the time to truly read through our manuscript. We feel that your comments were very helpful for increasing the quality of the paper to its current level. Your comments, together with those of referee #2, led to a thorough revision of the paper.

5   The most general comments regarding the revisions to the manuscript are:

1. Due to the maize cropland at Yangling station was irrigated and the maize was not suffered severe water and heat stress, a grassland, which experienced severe drought at Vaira Ranch (US-Var) Fluxnet site, was used to validate the ability of STEMMUS-SCOPE responding to drought.

2. Some figures and tables were changed. We updated Table 1 because the previous table ignored the latest improvements
10      in some LSMs. We added the Table 2 for presenting the difference among SCOPE, SCOPE_SM, STEMMUS, and STEMMUS-SCOPE. The figure of root length density was changed into a table for comparing simulated root length density and observed that in different sites.

3. We presented modeled half-hourly transpiration, gross primary production, solar-induced fluorescence, and leaf water potential and analyzed the relationship among gross primary production, solar-induced fluorescence, and leaf water
15      potential.

4. We added a new section for discussing the advanced plant hydraulics and root growth processes and what STEMMUS-SCOPE needs to be improved in the next step. Besides, we have tried to enhance the structure in the revised manuscript.

The reviewers' comments are in black and our responses in blue.

Main concerns:

20  ***Comments 1****: SCOPE is a vertical (1-D) integrated radiative transfer and energy balance model, which is widely used in the simulations of vegetation photosynthesis process and fluorescence at the leaf and canopy level. The soil model is very simplified in SCOPE. It is interesting to see the STEMMUS Model, which is good at dealing with the mass and heat transfer processes in unsaturated soil, is implemented into SCOPE. However, I do not see significant improvements in the SCOPE_STEMMUS model in current manuscript, although the SCOPE_STEMMUS includes root water uptake in unsaturated soil. I think this*
25  *shortcoming lies in the model validations based on the measurements at the Yangling station. As we know, SCOPE model has*

*the abilities to simulate the vegetation photosynthesis and evapotranspiration under the unstressed water conditions (Zhang et al., 2020; Shan et al., 2019; Zhang et al., 2018). And the Yangling station has irrigation and vegetation growth do not have water and heat stresses in 2017. Therefore, we can see very similar simulations from SCOPE and SCOPE_STEMMUS in Figure 6 and 7, which means the similar ability of SCOPE_STEMMUS and SCOPE in simulating ET and T. I think more sites*
30 *that have water or heat stresses should be used for the validations to prove the better ability of SCOPE_STEMMUS.*

**Response:** Thanks for the constructive comment! The authors added a new validation at a grassland which was the same FLUXNET site used in Bayat's paper. The advantage of STEMMUS-SCOPE can be seen obviously in the simulation at grassland, especially when the vegetation experiencing moderate water stress. The reason is that STEMMUS-SCOPE considered root length density and soil water content distribution in the root zone. However, SCOPE_SM could overestimate
35 or underestimate the effect of water stress due to it only using the soil water content at a specific soil depth or average root zone soil moisture. For the grassland, when the dry season coming, the surface soil was very dry, and grass root can absorb deep soil water to meet the high transpiration rate. So, it is not reasonable using the soil water content at 10 cm depth to calculate water stress factor. For the maize cropland, as we use the averaged root zone soil water content as the input of SCOPE_SM, the model will underestimate the effect of drought. The reason is that the maize root was concentrated at 20 cm
40 to 40cm soil depth. The higher water content in deep soil cannot be fully used by maize in SCOPE-SM because of the less root distribution in deep soil. Furthermore, the authors are working on validating the coupled model at more different ecosystem. The current paper is focused on model development (Page 28, line 319-325; Page 32, line 380-388)

***Comments 2**: For the development of SCOPE model, Bayat et al. (2019) have extended the SCOPE model to combine optical reflectance and soil moisture observations for remote sensing of ecosystem functioning under water stress conditions. Bayat's*
45 *work has overcome the shortcoming in biased estimations of GPP and ET under water stressed conditions and the significant improvements of GPP and ET in SCOPE_SM model have shown in the paper. We also see the same abilities of SCOPE_SM and SCOPE_STEMMUS in simulating ET in this manuscript. Therefore, the authors should declare what the improvements are in SCOPE_STEMMUS model. Terrestrial biosphere models typically use empirical functions to represent vegetation responses to soil drought, especially in the water-limited areas. These functions largely neglect recent advances in plant*
50 *ecophysiology that link xylem hydraulic functioning with stomatal responses to climate. I think this may be a direction to declare the new insights in the impacts of water tress on the vegetation growth, due to the descriptions of root water uptake in STEMMUS model.*

**Response:** Although Bayat et al. (2019) have extended the SCOPE model to combine optical reflectance and soil moisture observations for remote sensing of ecosystem functioning under water stress conditions, the distribution of fine root and soil moisture were ignored. For the very wet or very dry condition, the soil moisture at a specific depth can not reflect the water content in the whole root zone and the root water uptake was not sensitive to root distribution. But when the vegetation suffering moderate water stress, the hydraulic redistribution (RH) process and compensatory root water uptake (CRWU) process enable the plant absorb more water in the (deeper) soil layer with high soil water content, which were not taken account in SCOPE_SM. These two processes were sensitive to vertical distribution of root system and soil moisture. These considerations enabled STEMMUS-SCOPE perform better when the grass site transited from wet season to dry season. Therefore, the coupled model accurately characterized the effect of moderate stress. The model can also be easily extended to include more plant hydraulics related plant traits but is beyond the scope of the current paper (Page 32, line 380-388)

*Comments 3: In Table 1, some information should be updated. Nowadays, CLM5.0, CALBLE and JULES have large improvements in the hydraulic functioning with stomatal responses to the warming climate (De Kauwe et al., 2020; Lawrence et al., 2020; Eller et al., 2020). And the authors should have more discussion about the root water uptake and the hydraulic functioning in the SCOPE_STEMMUS model in this manuscript.*

**Response:** Thank you for sharing these useful references! As this study was conducted in 2019, some update in these LSMs was not included in the previous manuscript. In this version, we updated the latest improvements in CLM5.0, CALBLE, Noah-MP, and JULES (Table 1). In addition, some discussion about hydraulic function was added (Section 3.8: Page 39, line 482-518).

*Comments 4: The quality of some figures should be improved. This paper focus on the model developments and the better ability of the new model should be clear to the readers. For example, Figure 2 should be removed to the supplemental material. And Figure 5 and 8 are difficult for the readers to see and these figures should be redraw.*

**Response:** Figure 2 was changed and the data description can be found in references. Besides, figure 5 and 8 were redrawn.

**Response to reviewer #2**

Thank you for the comprehensive comments. Your comments, together with those of referee #1, led to a thorough revision of the paper.

The most general comments regarding the revisions to the manuscript are:

1. Due to the maize cropland at Yangling station was irrigated and the maize was not suffered severe water and heat stress, a grassland, which experienced severe drought at Vaira Ranch (US-Var) Fluxnet site, was used to validate the ability of STEMMUS-SCOPE responding to drought.

2. Some figures and tables were changed. We updated Table 1 because the previous table ignored the latest improvements in some LSMs. We added the Table 2 for presenting the difference among SCOPE, SCOPE_SM, STEMMUS, and STEMMUS-SCOPE. The figure of root length density was changed into a table for comparing simulated root length density and observed that in different sites.

3. We presented modeled half-hourly transpiration, gross primary production, solar-induced fluorescence, and leaf water potential and analyzed the relationship among gross primary production, solar-induced fluorescence, and leaf water potential.

4. We added a new section for discussing the advanced plant hydraulics and root growth processes and what STEMMUS-SCOPE needs to be improved in the next step. Besides, we have tried to enhance the structure in the revised manuscript.

The reviewers' comments are in black and our responses in blue.

Main concerns:

***Comments 1****: It was difficult to see what are overarching scientific question and findings (including development of a novel model) in the current manuscript. Although the authors mentioned "most of the current vegetation photosynthesis models do not account for root water uptake, which compromises their applications under water, stressed conditions (P1L15-)", it should be noted that there are numerous SPAC models that are successful in taking into consideration the root water uptake (the authors should look at the pioneer paper (Williams et al. 1996, PCE 19, 911-927)). I think all figures shown in this manuscript can be reproduced by most existing SPAC models including most DGVMs, and thus, I feel they are meaningless to be represented.*

100     *Frankly speaking, because of the above reason I feel the current manuscript cannot be reviewed anymore, but I also feel this work is very potential. I acknowledge SCOPE has a huge advantage in terms of calculation of leaf-scale chlorophyll fluorescence. Thus, as the authors mentioned at the end of the manuscript (P21L392-), SCOPE_STEMMUS can be very state-of-the-arts SPAC model that can simulate the effect of plant water stress via soil moisture status on leaf-to-canopy scale chlorophyll fluorescence.*

105     *Thus, I will reject the current manuscript temporarily, but I strongly encourage the authors to resubmit this work with adding modelling results and discussion about the effect of plant water stress via soil moisture status on leaf-to-canopy scale chlorophyll fluorescence, which might be easily simulated using SCOPE_STEMMUS. For this, the authors should note: Obviously SCOPE_STEMMUS failed to reproduce the root developments (Fig. 12), but is successful in reproduction of transpiration and NEE. This is a serious inconsistency that prevents sound simulations of the effect of water stress on leaf gas*
110     *exchange, and must be solved for resubmitting this work.*

    **Response:** Indeed, SCOPE has a huge advantage in terms of calculation of leaf-scale chlorophyll fluorescence (*SIF*). We have added and compared the simulated *SIF* of SCOPE, SCOPE_SM, and STEMMUS-SCOPE, and analyzed the relationship between *SIF* and leaf water potential (*LWP*). In addition, the authors are sorry for not presenting the simulation of root length density clearly in the previous manuscript. Actually, STEMMUS-SCOPE well simulated root growth. The simulated root
115     length density (*RLD*) was comparable to the measurements from sites from Beijing, China, Potenza, Italy, and Tokyo, Japan. These lines were shaded by the simulated *RLD* by STEMMUS-SCOPE in the previous figure. In the revised manuscript, the authors changed Figure 12 into a table which can present simulated *RLD* more clearly. (Page 37, line 448-474)

    **Comments 2**: *Though this is trivial point compared to the above-mentioned, I assumed the first author is an inexperienced scientist. For example, there was an ambiguous definition between "Results" and "Discussion" sections and were many wrong*
120     *wordings. So I recommend to resubmit your paper to academic journals after thorough checking by the other experienced authors.*

    **Response:** Thanks for your comment. The manuscript has been reconstructed and modified by senior authors.

**Response to executive editor**

*In my role as Executive editor of GMD, I would like to bring to your attention our Editorial version 1.2:*

125     *https://www.geosci-model-dev.net/12/2215/2019/ This highlights some requirements of papers published in GMD, which is*

*also available on the GMD website in the 'Manuscript Types' section:*

*http://www.geoscientific-model-development.net/submission/manuscript_types.html*

*In particular, please note that for your paper, the following requirements have not been met in the Discussions paper:*

*• "The main paper must give the model name and version number (or other unique identifier) in the title."*

130     *• "If the model development relates to a single model then the model name and the version number must be included in the*

*title of the paper. If the main intention of an article is to make a general (i.e. model independent) statement about the usefulness*

*of a new development, but the usefulness is shown with the help of one specific model, the model name and version number*

*must be stated in the title. The title could have a form such as, "Title outlining amazing generic advance: a case study with*

*Model XXX (version Y)"."*

135     *As SCOPE is the model to which all your modifications and evaluation applies, SCOPE and a unique identifier, for the*

*resulting version after integration of your modifications, should be named in the title of your article in your revised submission*

*to GMD.*

**Response:** Thanks, Astrid Kerkweg, for your comments! We added the name of the coupled model to the title in this revised manuscript.

[revised manuscript text omitted]

---

## Author Comment (AC2) · 23 Nov 2020

Thank you for the comprehensive comments. Your comments, together with those of referee #1, led to a thorough revision of the paper.

Please also note the supplement to this comment:
https://gmd.copernicus.org/preprints/gmd-2020-85/gmd-2020-85-AC2-supplement.pdf
* * *

---

## Author Comment (AC3) · 23 Nov 2020

Thanks, Astrid Kerkweg, for your comments! We added the name of the coupled model to the title in this revised manuscript.

Please also note the supplement to this comment:
https://gmd.copernicus.org/preprints/gmd-2020-85/gmd-2020-85-AC3-supplement.pdf

---

## Author Comment (AC4) · 23 Nov 2020

Thank you for the comprehensive comments, and also for taking the time to truly read through our manuscript. We feel that your comments were very helpful for increasing the quality of the paper to its current level. Your comments, together with those of referee #2, led to a thorough revision of the paper.

Our responses are in the attached PDF file.

Please also note the supplement to this comment:
https://gmd.copernicus.org/preprints/gmd-2020-85/gmd-2020-85-AC4-supplement.pdf

---

## Author Comment (AC6) · 23 Nov 2020

Thanks, Hisashi Sato, for your recommendation! The authors have tried their best to improve the manuscript and submitted the final response.

Please also note the supplement to this comment:
https://gmd.copernicus.org/preprints/gmd-2020-85/gmd-2020-85-AC6-supplement.pdf

---

## Author Response (AR2)

**Response to reviewer #3**

Thank you for the comprehensive comments, and also for taking the time to truly read through our manuscript. We feel that your comments were very helpful for increasing the quality of the paper to its current level.

The reviewer's comments are in black and our responses in blue.

5    Main concerns:

*Comments 1: There is lack of validation of vertical root distribution and seasonal root length variations based on evidence of in situ observation. It is necessary to clarify the limitation of model and add some reliable proof for the root growth modeling or remaining future work for model development.*

**Response:** Thanks for your comment! Indeed, this study does not validate the simulation of seasonal vertical root length profile

10    due to lacking of in situ observations. We are trying to collect some filed measured maize root length density to validate the root growth simulation ability of STEMMUS-SCOPE, and some relevant results will be published in a future work. (line 451-452)

Details:

*Comments 1: L42:Oxygen for plant is not main topic of this paper. Please remove "oxygen".*

15    **Response:** Done. (line 42)

*Comments 2: L76:Please show the simple introduction of STEMMUS itself, and explain why STEMMUS is necessary to determine RWU based on detail process in STEMMUS model (compared to SPAC models as reviewer#2 mentioned). Please also remined the importance of heat transfer processes which is not uncovered through the Introduction.*

**Response:** Thanks for your comment! We have added a simple description of STEMMUS, more detailed information about

20    STEMMUS can be found in the section of methodology and data. Previous SPAC simplified the water and heat transfer process and the soil layers was fixed. In STEMMUS, the soil layer can be set flexible. In addition, the heat transfer (the soil temperature) is vital for vegetation phenology development and freeze-thaw processes. (line 79, line 121 to 123)

*Comments 3: L63, L67, L69:Vegetation appearance may be replaced to remote sensing signals. It sounds ambiguous terms.*

**Response:** Done. (line 63, line 67, line 68)

25      ***Comments 4****: L82: Please add "SIF" in the purpose as one of Results chapter focused on SIF, and that is the advantage of SCOPE.*

**Response:** Done. (line 84)

***Comments 5****: L103: Please add "SIF" in the sentence, as same manner in the chapter of the Results. Please add clear explanation of geometry parameter on SIF at nadir or any view zenith angle or field of view angle.*

30      **Response:** Done. (line 103)

***Comments 6****: L122−123: Please move the sentence to Introduction.*

**Response:** We request to keep it here and we have added a relevant explanation in the Introduction section. (line 80-81)

***Comments 7****: L135: Please add version information of SCOPE which you based on for the coupling model.*

**Response:** Done. (line 139)

35      ***Comments 8****: L137: Please add "SIF" among calculated output on SCOPE coupling model.*

**Response:** Done. (line 141)

***Comments 9****: L175: Please clearly show leaf water potential and water stress factor in here and in Table2, section 3.6 as well.*

**Response:** Done. (line 179-180)

***Comments 10****: L201: Please add general description of exact values of RMSE and d as same manner in next section.*

40      **Response:** Done. (line 201-204)

***Comments 11****: L252: There are mismatch of drought water stressed period as follows. DOY193-202 for ET in Yangling. L268: DOY183-202 for transpiration in Yangling. L346: DOY192-202 for WSF in Yangling.*

**Response:** Done.

***Comments 12****: L270: "better simulation" is not clear description. Please correct a little more detail.*

45      **Response:** Done. (line 278)

***Comments 13****: L328: Please move the following sentence to the method "Leaf water potential is a parameter to reflect plant water status."*

**Response:** Done. (line 144-145)

*Comments 14*: L354: Table 3 did not contain root distribution gradient information. The Author also did not clearly present the validation of root depth between model prediction and observation in maize and grassland species. Before referring previous studies, the quality of root depth representation should be shown in proper manners.

**Response:** Relevant sentence has been corrected. (line 364)

*Comments 15*: L337: Martineau et al (2017) descripted leaf water potential in MPa pressure unit. Please refer the original values too. And please show the proof of unit conversion.

**Response:** Thanks for your comment! We have added the original values in the reference and showed how does the unit converted. (line 345-346)

*Comments 16*: L388: Please check correct figure number (Figure 13a.).

**Response:** Done. (line 396)

*Comments 17*: L394: Please check correct figure number (Figure 13a.).

**Response:** Done. (line 403)

*Comments 18*: L395: Please explain the reason why you chose DOY199. You may show the validity that SIF-drought response was not affected by dependence on PAR variations in diurnal cycle. (e.g. sky clearness stability during daytime)

**Response:** Thanks for your comment! We have added the reason why the DOY 199 was selected. (line 401-402)

*Comments 19*: L418: This sentence is unclear. Is there any evidence that embolism and refilling process cannot occur in cropland and grassland? Please make clear that is related to whether land use, meteorology or plant traits.

**Response:** Thanks for your comment! The threshold which can result in embolism of maize is about -1.6 MPa (Cochard, 2002), and the maize cropland in Yangling was not reached such a low leaf water potential. For the grassland, it experienced a very severe dry season and the grass was dead or dormant. So, this study cannot validate the effect of embolism and refilling on the drought response ability of STEMMUS-SCOPE.

*Comments 20*: L437: "though it well predicted root depth which is the most critical factor" Actual validation of the root depth prediction was not found in the manuscript.

**Response:** It has been corrected. (line 445)

*Comments 21*: L455: As you noted in L464, the study did not validate modeled SIF against observations. It may be replaced as potential applicability.

**Response:** It has been revised. (line 466)

*Comments 22*: L465: *"an effective observation operator to simulate remote sensing signals" It is not clear what you want to say. Please revise plainer sentence.*

**Response:** Done. (line 475)

*Comments 23*: L593: *Soil water potential and leaf water potential ψleaf are noted as pressure head unit (head, m) in this study. Please describe the unit of water potential clearly, to avoid confusing to MPa pressure unit.*

80

**Response:** The unit of water potential has been described in detail in the manuscript. (Line 602)

**References**

Cochard, H. (2002). Xylem embolism and drought-induced stomatal closure in maize. Planta Berlin.